# TORCHTITAN: ONE-STOP PYTORCH NATIVE SOLUTION FOR PRODUCTION READY LLM PRETRAINING

**Wanchao Liang**[1], **Tianyu Liu**[1]\*, **Less Wright**[1], **Will Constable**[1], **Andrew Gu**[1]
**Chien-Chin Huang**[1], **Iris Zhang**[1], **Wei Feng**[1], **Howard Huang**[1], **Junjie Wang**[1]
**Sanket Purandare**[2][†], **Gokul Nadathur**[1], **Stratos Idreos**[2]
[1]Meta, [2]Harvard University

## ABSTRACT

The development of large language models (LLMs) has been instrumental in advancing state-of-the-art natural language processing applications. Training LLMs with billions of parameters and trillions of tokens require sophisticated distributed systems that enable composing and comparing several state-of-the-art techniques in order to efficiently scale across thousands of accelerators. However, existing solutions are complex, scattered across multiple libraries/repositories, lack interoperability, and are cumbersome to maintain. Thus, curating and empirically comparing training recipes require non-trivial engineering effort.

This paper introduces TORCHTITAN, an open-source[1], PyTorch-native distributed training system that unifies and advances state-of-the-art techniques, streamlining integration and reducing engineering overhead. TORCHTITAN enables seamless application of 4D parallelism in a modular and composable manner, while featuring elastic scaling to adapt to changing computational requirements. The system provides comprehensive logging, efficient checkpointing, and debugging tools, ensuring production-ready training. Moreover, TORCHTITAN incorporates innovative hardware-software co-designed solutions, leveraging cutting-edge features like Float8 training and SymmetricMemory to maximize hardware utilization. As a flexible experimental test bed, TORCHTITAN facilitates the curation and comparison of custom recipes for diverse training contexts. By leveraging TORCHTITAN, we developed optimized training recipes for the Llama 3.1 family and provide actionable guidance on selecting and combining distributed training techniques to maximize training efficiency, based on our hands-on experiences.

We thoroughly assess TORCHTITAN on the Llama 3.1 family of LLMs, spanning 8 billion to 405 billion parameters, and showcase its exceptional performance, modular composability, and elastic scalability. By stacking training optimizations, we demonstrate accelerations ranging from 65.08% on Llama 3.1 8B at 128 GPU scale (1D), 12.59% on Llama 3.1 70B at 256 GPU scale (2D), to 30% on Llama 3.1 405B at 512 GPU scale (3D) on NVIDIA H100 GPUs over optimized baselines. We also demonstrate the effectiveness of 4D parallelism in enabling long context training.

## 1 INTRODUCTION

Large Language Models (LLMs) (Devlin, 2018; Liu et al., 2019; Radford et al., 2019; Chowdhery et al., 2023; Anil et al., 2023; Achiam et al., 2023; Dubey et al., 2024; Jiang et al., 2024; Abdin et al., 2024) have been the driving force behind the advancement of natural language processing (NLP) applications spanning language translation, content/code generation, conversational AI, text data analysis, creative writing and art, education, and research, etc.

Achieving state-of-the-art LLM performance requires massive scale, exemplified by top-performing models like Llama 3.1 (405B parameters, 15T tokens, 30.84M GPU hours, 16K H100 GPUs) (Dubey

---

\*Corresponding author: Tianyu Liu (`lty@meta.com`)

†Work done at Meta

[1]Github: https://github.com/pytorch/torchtitan

et al., 2024) and Google's PaLM (540B parameters, 0.8T tokens, 9.4M TPU hours, 6144 TPUv4 chips) (Chowdhery et al., 2023). These models demonstrate exceptional natural language understanding and generation capabilities, but at the same time necessitate substantial computational resources, memory, and time to train, highlighting the significant investment required to advance natural language processing.

Training large language models (LLMs) at scale is a daunting task that requires a delicate balance of parallelism, computation, and communication, all while navigating intricate memory and computation trade-offs. The massive resources required for training make it prone to GPU failures, underscoring the need for efficient recovery mechanisms and checkpointing strategies to minimize downtime (Eisenman et al., 2022; Wang et al., 2023; Gupta et al., 2024; Maurya et al., 2024; Wan et al., 2024). To optimize resource utilization and achieve elastic scalability, it is crucial to combine multiple parallelism techniques, including Data Parallel (Li et al., 2020; Rajbhandari et al., 2020; Zhang et al., 2022; Zhao et al., 2023), Tensor Parallel (Narayanan et al., 2021; Wang et al., 2022; Korthikanti et al., 2023), Context Parallel (Liu et al., 2023; Liu & Abbeel, 2024; NVIDIA, 2023; Fang & Zhao, 2024), and Pipeline Parallel (Huang et al., 2019; Narayanan et al., 2019; 2021; Qi et al., 2023). By stacking these parallelisms with memory and computation optimization techniques, such as activation recomputation (Chen et al., 2016; Korthikanti et al., 2023; He & Yu, 2023; Purandare et al., 2023), mixed precision training (Micikevicius et al., 2018; 2022), and deep learning compilers (Bradbury et al., 2018; Yu et al., 2023; Li et al., 2024; Ansel et al., 2024), it is possible to maximize hardware utilization.

While state-of-the-art distributed training techniques have significantly advanced the field, existing systems that incorporate them still fall short in addressing critical challenges that hinder their usability, adoption and effectiveness for researchers and industry practitioners.

1. *Non-composable*: Existing systems struggle to integrate and stack parallelism techniques, limiting multi-dimensional exploration and integration with memory and computation optimizations, thereby reducing training efficiency.

2. *Inflexible Architecture*: Lack of modularity and extensibility hampers the integration of new techniques, optimizations, and hardware, limiting adaptability to evolving ML landscapes.

3. *Inefficient Hardware Utilization*: Poor leverage of advanced hardware features results in sub-optimal GPU efficiency and lack of customizable checkpointing strategies for memory-computation trade-offs.

4. *Insufficient Support for Production Training*: Limited distributed checkpointing scalability, cumbersome failure recovery, and inadequate debugging tools hinder production-grade workflows.

5. *Framework Limitations*: Dependence on external, poorly maintained dependencies and failure to harness PyTorch's optimized kernels, new features, and compiler support lead to inefficiencies and compatibility issues.

The non-composability and inflexibility of distributed systems stem from the absence of unified tensor and device abstractions applied consistently across the stack. Without these foundational components, parallelism strategies, checkpointing, and efficiency optimizations remain fragmented, limiting modularity, scalability, and extensibility.

TORCHTITAN 's primary research contribution lies in identifying and unifying the core principles of parallelism and optimization techniques into a cohesive framework. By leveraging and extending PyTorch's Distributed Tensor (DTensor) and DeviceMesh (PyTorch Community, 2023a), TORCHTITAN provides a unified abstraction that simplifies the composition of parallelism strategies, and ensures correct single device semantics with its sharding primitives. Unlike existing systems that often rely on rigid or ad-hoc designs, TORCHTITAN introduces a unified template for distributed training, enabling researchers to systematically explore configurations, rigorously evaluate existing methods, and uncover novel techniques within the design space.

TORCHTITAN represents a complete distributed training system for large language models (LLMs), rather than merely a collection of individual techniques. Its modular, extensible architecture supports seamless composition of 4D parallelism, advanced training optimizations, and scalable distributed checkpoint save/load, all while harnessing PyTorch's native capabilities. The system not only en-

able production-grade training with thousands of GPUs, but also reduces complexity and fosters innovation, setting a new standard for scalable and flexible distributed training systems.

To develop and evaluate the capabilities of TORCHTITAN, we undertook several key steps, which represent the core contributions of this work, and are summarized as follows:

1. We advance DTensor by extending its sharding to support n-D parallelism, adding compatibility with `torch.compile` for compiler optimizations, and enabling efficient checkpointing of n-D models via state dict support. We also resolve critical bugs to bolster DTensor's production readiness.

2. We demonstrate how to compose various parallelism techniques, facilitating the exploration of multi-dimensional parallelism in large language model training (§2.1).

3. We enable novel hardware-software co-designed solutions exploiting advanced hardware features to increase GPU efficiency, offer customizable activation checkpointing strategies for navigating memory-computation trade-offs, and utilize `torch.compile` to further optimize memory, computation, and communication (§2.2).

4. We offer production grade training by incorporating scalable and efficient distributed checkpoint to facilitate fast failure recovery, integrating debugging tools like Flight Recorder to debug crashed/stuck jobs, and provide extensive logging metrics (§2.3).

5. We extensively evaluate TORCHTITAN on Llama 3.1 family of models, stacking 1D to 4D parallelisms (respectively), at the scale from 8 to 512 GPUs to demonstrate elastic scalability while ensuring efficiency, convergence, and accuracy. In summary, we demonstrate training accelerations ranging from 65.08% on Llama 3.1 8B at 128 GPU scale (1D), 12.59% on Llama3.1 70B at 256 GPU scale (2D), to 30% on Llama3.1 405B at 512 GPU scale (3D), and the effectiveness of 4D parallelism in enabling long context training, on latest NVIDIA H100 GPUs over optimized baselines (§3.2).

6. We provide systematic training recipes and guidelines that empower users to navigate the complexities of distributed training, helping them optimize training efficiency for a range of model sizes and cluster configurations (§3.3).

By providing an accessible and extensible platform, TORCHTITAN democratizes large language model (LLM) pretraining, empowering a wider range of researchers and developers to tap into the potential of LLMs and accelerate innovation in the field.

## 2 ELASTICITY THROUGH COMPOSABILITY

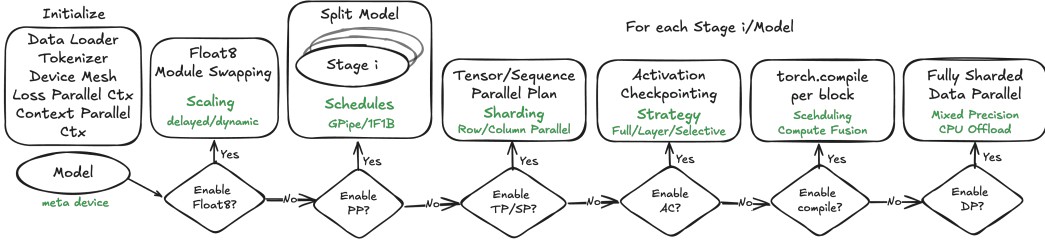

Figure 1: Composable and Modular TORCHTITAN initialization workflow.

TORCHTITAN incorporates various parallelisms in a modular manner to enable easy, user-selectable combinations of multi-dimensional shardings. This composability enables the tackling of difficult scaling challenges by enhancing the ease of exploration for optimizing training efficiencies at scale.

The codebase of TORCHTITAN is organized purposefully to enable composability and extensibility. We intentionally keep three main components separate and as orthogonal as possible: (1) the model definition, which is parallelism-agnostic and designed for readability, (2) parallelism helpers, which apply parallelisms and training optimizations to a particular model, and (3) a generalized training loop. All these components are configurable via TOML files with command-line overrides, and it is easy to add new models and parallelism techniques on top of the existing codebase.

## 2.1 COMPOSABLE N-D PARALLELISM TRAINING

In this section, we will walk through the entire regime of scaling model training on large clusters, including meta device initialization and the core composable multi-dimensional parallelisms, to showcase how these techniques can be composed to train LLMs efficiently at increasing scale in TORCHTITAN. The corresponding code snippets in TORCHTITAN can be found in Appendix A.

### 2.1.1 LARGE-SCALE MODEL INITIALIZATION USING META DEVICE

As LLMs grow exponentially, scaling challenges arise even before training begins, particularly in instantiating large models for sharding without exceeding CPU or GPU memory limits.

To address this, TORCHTITAN enables meta device initialization, where the model is first created on a *meta* device that stores only metadata, making initialization ultra-fast. The model is then sharded into Distributed Tensors (DTensors), with the local shard of each parameter residing on the meta device. Finally, parameter initialization is performed using user-defined functions, ensuring correct DTensor sharding layouts and proper RNG seed usage.

### 2.1.2 FULLY SHARDED DATA PARALLEL

The original Fully Sharded Data Parallel (FSDP) (Zhao et al., 2023) is an effective implementation of ZeRO that offers large model training capability in PyTorch. However, the original implementation (FSDP1) in PyTorch suffers from various limitations due to its FlatParameter implementation.

Given these limitations, TORCHTITAN integrates a new version of Fully Sharded Data Parallel (FSDP2), which uses the per-parameter Distributed Tensor sharding representation and thus provides better composability with model parallelism techniques and other features that require the manipulation of individual parameters.

TORCHTITAN integrates and leverages FSDP2 as it's default 1D parallelism, benefiting from the improved memory management (often 7 percent lower per GPU memory requirement vs FSDP1) and the slight performance gains (average of 1.5 percent gain vs FSDP1). More details on FSDP2 and usage example are shown in Appendix B.1. TORCHTITAN makes it simple to run with FSDP2 by embedding appropriate defaults, including auto-sharding with your world size automatically.

For scaling to even larger world sizes, TORCHTITAN also integrates Hybrid Sharded Data Parallel (HSDP) which extends FSDP2 by creating 2D DeviceMesh with replica groups. Details are shown in Appendix B.2

### 2.1.3 TENSOR PARALLEL

Tensor Parallel (TP) (Narayanan et al., 2021), together with Sequence Parallel (SP) (Korthikanti et al., 2023), is a key model parallelism technique to enable large model training at scale.

TP is implemented in TORCHTITAN using the PyTorch's `RowwiseParallel` and `ColwiseParallel` APIs, where the model parameters are partitioned to DTensors and perform sharded computation with it (Figure 3). By leveraging DTensor, the TP implementation does not need to touch the model code, which allows faster enablement on different models and provides better composability with other features mentioned in this paper.

**Tensor and Sequence Parallel (TP/SP)**   While TP partitions the most computationally demanding aspects, Sequence Parallel (SP) performs a sharded computation for the normalization or dropout layers on the sequence dimension, which otherwise generate large replicated activation tensors, and thus can be challenging to memory constraints per GPU. See Appendix B.3 for more details, illustrations, and usage for both TP and FSDP + TP.

Due to the synergistic relationship between TP and SP, TORCHTITAN natively bundles these two together, and they are jointly controlled by the TP degree setting.

**Loss Parallel**   When computing the loss function, model outputs are typically large, especially with TP/SP, where they are sharded across the vocabulary dimension. Naively computing cross-entropy loss requires gathering all shards, leading to high memory usage.

Loss Parallel enables efficient loss computation without fully gathering model outputs, significantly reducing memory consumption and improving training speed by minimizing communication overhead and enabling parallel sharded computation. Due to these advantages, TORCHTITAN implements Loss Parallel by default.

### 2.1.4 PIPELINE PARALLEL

For large-scale pretraining, TORCHTITAN employs Pipeline Parallelism (PP), which minimizes communication overhead by leveraging P2P communications. PP divides the model into $S$ stages, each running on a separate group of devices. Typically, each stage represents a model layer or a group of adjacent layers, but can include partial layers. During the forward pass, each stage receives input activations (except stage 0), computes locally, and sends output activations (except stage $S - 1$). The last stage computes the loss and initiates the backward pass, sending gradients in reverse order. To improve efficiency, the input batch is split into microbatches, and the pipeline schedule overlaps computation and communication across microbatches. TORCHTITAN supports various pipeline schedules (Narayanan et al., 2019; Huang et al., 2019; Narayanan et al., 2021; Qi et al., 2023). Recently, TORCHTITAN added support for new schedules including ZeroBubble and 'Flexible-Interleaved-1F1B', making use of pipeline IR to quickly express new schedules as a list of compute actions and rely on compiler passes to insert and optimize communication actions PyTorch Team 2024d.

The PP training loop differs from standard training by creating pipeline stages and executing schedules instead of directly invoking `model.forward()`. Since loss is computed per microbatch, TORCHTITAN introduces a shared `loss_fn` to unify pipeline and non-pipeline workflows, reducing code divergence.

`torch.distributed.pipelining` also simplifies interactions with data parallelism, ensuring that reductions occur only after the final microbatch and handling shard/unshard operations (e.g., with ZeRO-3), as well as applying gradient scaling transparently within the pipeline schedule executor. For more details on TORCHTITAN's implementation of PP, see Appendix B.4.

### 2.1.5 CONTEXT PARALLELISM

TORCHTITAN has been extended to incorporate Context Parallelism (CP) (Liu et al., 2023; Liu & Abbeel, 2024; NVIDIA, 2023), enabling 4D parallelism by adding CP as an additional dimension to existing DP, TP, and PP. CP scales model training by splitting the context dimension across GPUs, significantly increasing the maximum trainable context length without causing out-of-memory (OOM) errors. For example, on Llama 3.1 8B with 8 H100 GPUs, using CP enabled training at context lengths up to 262,144 tokens, achieving minor MFU degradation as CP degree increases (PyTorch Team, 2025). For more details on CP integration please refer to Appendix B.5.

## 2.2 OPTIMIZING TRAINING EFFICIENCIES

### 2.2.1 NAVIGATING COMPUTE-MEMORY TRADE-OFFS USING ACTIVATION CHECKPOINTING

Activation checkpointing (AC) (Chen et al., 2016; He & Yu, 2023; Purandare et al., 2023) and selective activation checkpointing (SAC) (Korthikanti et al., 2023) are standard training techniques to reduce peak GPU memory usage, by trading activation recomputation during the backward pass for memory savings. It is often needed even after applying multi-dimensional parallelisms.

TORCHTITAN offers flexible AC and SAC options utilizing `torch.utils.checkpoint`, applied at the `TransformerBlock` level. The AC strategies include "full" AC, op-level SAC, and layer-level SAC.

Within a `TransformerBlock`, full AC works by recomputing all activation tensors needed during the backward pass, whereas op-level SAC saves the results from computation-intensive PyTorch operations and only recomputes others. Layer-level SAC works in similar fashion as full AC, but the wrapping is applied to every $x$ `TransformerBlock` (where $x$ is specified by the user) to implement configurable trade-offs between memory and recompute. (Details are in Appendix B.6.)

### 2.2.2 REGIONAL COMPILATION TO EXPLOIT `TORCH.COMPILE` OPTIMIZATIONS

`torch.compile` was released in PyTorch 2 (Ansel et al., 2024) with TorchDynamo as the frontend to extract PyTorch operations into an FX graph, and TorchInductor as the backend to compile the FX graph into fused Triton code to improve the performance.

In TORCHTITAN, we use regional compilation, which applies `torch.compile` to each individual `TransformerBlock` in the Transformer model. This has two main benefits: (1) we get a full graph (without graph breaks) for each region, compatible with FSDP2 and TP (and more generally `torch.Tensor` subclasses such as DTensor) and other PyTorch distributed training techniques; (2) since the Llama model stacks identical `TransformerBlock` layers one after another, `torch.compile` can identify the same structure is being repeatedly compiled and only compile once, thus greatly reducing compilation time.

`torch.compile` brings efficiency in both throughput and memory (see Section 3.2) via computation fusions and computation-communication reordering, in a model-agnostic way with a simple user interface. Below we further elaborate how `torch.compile` composability helps TORCHTITAN unlock hardware-optimized performance gain with simple user interface, with the integration of advanced features such as Asynchronous TP and Float8.

### 2.2.3 ASYNCHRONOUS TENSOR PARALLEL TO MAXIMALLY OVERLAP COMMUNICATION

By default, TP incurs blocking communications before/after the sharded computations, causing computation resources to not be effectively utilized. Asynchronous TP (AsyncTP) (Wang et al., 2022) achieves computation-communication overlap by fractionalizing the TP matrix multiplications within attention and feed-forward modules into smaller chunks, and overlapping communication collectives in between each section. The overlap is achieved by a micro-pipelining optimization, where results are being communicated at the same time that the other chunks of the matmul are being computed.

PyTorch AsyncTP is based on a `SymmetricMemory` abstraction, which creates intra-node buffers to write faster communication collectives. This is done by allocating a shared memory buffer on each GPU in order to provide direct P2P access (PyTorch Team, 2024a).

With TORCHTITAN's integration of `torch.compile`, AsyncTP can be easily configured in TORCHTITAN to achieve meaningful end-to-end speedups (see Section 3.2 for details) on newer hardware (H100 or newer GPUs with NVSwitch within a node). Usage details are in Appendix B.7

### 2.2.4 BOOSTING THROUGHPUT WITH MIXED PRECISION TRAINING AND FLOAT8 SUPPORT

Mixed precision training (Micikevicius et al., 2018) provides both memory and computational savings while ensuring training stability. FSDP2 has built-in support for mixed precision training with basic `torch.dtype`. This covers the popular usage of performing FSDP all-gather and computation in a low precision (e.g. `torch.bfloat16`), and perform lossless FSDP reduce-scatter (gradient) in high precision (e.g. `torch.float32`) for better numerical results. See Appendix B.8 for usage details.

TORCHTITAN also supports more advanced mixed precision training with Float8, a derived data type, applied selectively to linear layers (available on newer hardware like NVIDIA H100), achieving substantial performance gains while ensuring training stability (reported in Section 3.2). The Float8 feature from `torchao.float8` supports multiple per-tensor scaling strategies, including dynamic, delayed, and static (see Micikevicius et al. (2022); PyTorch Community (2023b), Section 4.3 for details), while being composable with other key PyTorch-native systems such as autograd, `torch.compile`, FSDP2 and TP (with Float8 all-gather capability) (PyTorch Team, 2024c).

### 2.3 PRODUCTION READY TRAINING

To enable production-grade training, TORCHTITAN offers seamless integration with key features out of the box. These include (1) efficient checkpointing using PyTorch Distributed Checkpointing (DCP), and (2) debugging stuck or crashed jobs through integration with Flight Recorder.

### 2.3.1 SCALABLE AND EFFICIENT DISTRIBUTED CHECKPOINTING

Checkpoints are crucial in training large language models for two reasons: they facilitate model reuse in applications like inference and evaluation, and they provide a recovery mechanism in case of failures. An optimal checkpointing workflow should ensure ease of reuse across different parallelisms and maintain high performance without slowing down training. There are two typical checkpointing methods. The first aggregates the state (model parameters and optimizer states) into an unsharded version that is parallelism-agnostic, facilitating easy reuse but requiring expensive communication. The second method has each trainer save its local sharded state, which speeds up the process but complicates reuse due to embedded parallelism information.

DCP addresses these challenges using DTensor, which encapsulates both global and local tensor information independently of parallelism. DCP converts this information into an internal format for storage. During loading, DCP matches the stored shards with the current DTensor-based model parameters and optimizer states, fetching the necessary shard from storage. TORCHTITAN effectively uses DCP to balance efficiency and usability. Furthermore, DCP enhances efficiency through asynchronous checkpointing by processing storage persistence in a separate thread, allowing this operation to overlap with subsequent training iterations. TORCHTITAN utilizes DCP's asynchronous checkpointing to reduce the checkpointing overhead by 5-15x compared to synchronous distributed checkpointing for the Llama 3.1 8B model (PyTorch Team, 2024b).

### 2.3.2 FLIGHT RECORDER TO DEBUG JOB CRASHES

Debugging NCCL collective timeouts at large scales is challenging due to the asynchronous nature of communication kernels. PyTorch's Flight Recorder addresses this by logging the start, end, and enqueue times for all collective and p2p operations, along with metadata like process groups, source/destination ranks, tensor sizes, and stack traces.

This data is invaluable for diagnosing hangs in parallelism code. For PP, it can pinpoint the latest send or recv completed on the GPU, helping debug schedule bugs. For FSDP and TP, it identifies ranks that failed to call collectives, aiding in uncovering issues with PP scheduling or TP logic.

## 3 EXPERIMENTATION

In this section, we demonstrate the effectiveness of elastic distributed training using TORCHTITAN, via experiments on Llama 3.1 8B, 70B, and 405B, from 1D parallelism to 4D parallelism, at the scale from 8 GPUs to 512 GPUs. We also share the knowledge and experience gained through TORCHTITAN experimentation. A walkthrough of the codebase on how we apply (up to) 4D parallelism can be found in Appendix A.

### 3.1 EXPERIMENTAL SETUP

The experiments are conducted on NVIDIA H100 GPUs[2] with 95 GiB memory, where each host is equipped with 8 GPUs and NVSwitch. Two hosts form a rack connected to a TOR switch. A back-end RDMA network connects the TOR switches. In TORCHTITAN we integrate a checkpointable data loader and provide built-in support for the C4 dataset (en variant), a colossal, cleaned version of Common Crawl's web crawl corpus (Raffel et al., 2020). We use the same dataset for all experiments in this section. For the tokenizer, we use the official one (tiktoken) released together with Llama 3.1.

### 3.2 PERFORMANCE

To showcase the elasticity and scalability of TORCHTITAN, we experiment on a wide range of GPU scales (from 8 to 512), as the underlying model size increases (8B, 70B, and 405B) with a varying number of parallelism dimensions (up to 4D). To demonstrate the effectiveness of the optimization techniques introduced in Section 2.2, we show how training throughput improves when adding each

---

[2]The H100 GPUs used for the experiments are non-standard. They have HBM2e and are limited to a lower TDP. The actual peak TFLOPs should be between SXM and NVL, and we don't know the exact value.

individual technique on appropriate baselines. In particular, when training on a higher dimensional parallelism with new features, the baseline is always updated to include all previous techniques.

We note that, throughout our experimentation, memory readings are stable across the whole training process[3], whereas throughput numbers (token per second, per GPU) are calculated and logged every 10 iterations, and always read at the (arbitrarily determined) 90th iteration. We do not report Model FLOPS Utilization (MFU) (Chowdhery et al., 2023) because when Float8 is enabled in TORCHTI-TAN, both BFLOAT16 Tensor Core and FP8 Tensor Core are involved in model training, but they have different peak FLOPS and the definition of MFU under such scenario is not well-defined. We note that the 1D Llama 3.1 8B model training on 8 or 128 H100 GPUs without Float8 achieves 33% to 42% MFU.

Table 1: 1D parallelism (FSDP) on Llama 3.1 8B model, 8 GPUs. Mixed precision training. Selective activation checkpointing. Local batch size 2, global batch size 16. (Stats per GPU)

| Techniques | Throughput (Tok/Sec) | Comparison | Memory (GiB) |
|---|---|---|---|
| FSDP | 6,258 | 100% | 81.9 |
| + `torch.compile` | 6,674 | + 6.64% | 77.0 |
| + `torch.compile` + Float8 | 9,409 | + 50.35% | 76.8 |

Table 2: 1D parallelism (FSDP) on Llama 3.1 8B model, 128 GPUs. Mixed precision training. Selective activation checkpointing. Local batch size 2, global batch size 256. (Stats per GPU)

| Techniques | Throughput (Tok/Sec) | Comparison | Memory (GiB) |
|---|---|---|---|
| FSDP | 5,645 | 100% | 67.0 |
| + `torch.compile` | 6,482 | + 14.82% | 62.1 |
| + `torch.compile` + Float8 | 9,319 | + 65.08% | 61.8 |

Table 3: 2D parallelism (FSDP + TP) + `torch.compile` + Float8 on Llama 3.1 70B model, 256 GPUs. Mixed precision training. Full activation checkpointing. FSDP degree 32, TP degree 8. Local batch size 16, global batch size 512. (Stats per GPU)

| Techniques | Throughput (Tok/Sec) | Comparison | Memory (GiB) |
|---|---|---|---|
| 2D | 897 | 100% | 70.3 |
| + AsyncTP | 1,010 | + 12.59% | 67.7 |

Table 4: 3D parallelism (FSDP + TP + PP) + `torch.compile` + Float8 + AsyncTP on Llama 3.1 405B model, 512 GPUs. Mixed precision training. Full activation checkpointing. FSDP degree 4, TP degree 8, PP degree 16. Local batch size 32, global batch size 128. (Stats per GPU)

| Schedule | Throughput (Tok/Sec) | Comparison | Memory (GiB) |
|---|---|---|---|
| 1F1B | 100 | 100% | 78.0 |
| Interleaved 1F1B | 130 | + 30.00% | 80.3 |

Additional experimental details and loss-convergence tests for correctness can be found in Appendix B.10.

### 3.3 SCALING WITH TORCHTITAN 4D PARALLELISM

Scaling large language models (LLMs) requires parallelism strategies to handle increasing model sizes and data on thousands of GPUs. TORCHTITAN enables efficient scaling through composable

---

[3]Different PP ranks can have different peak memory usages. We take the maximum across all GPUs.

Table 5: FSDP + CP + `torch.compile` + Float8 on Llama 3.1 8B model, 8 GPUs. Mixed precision training. Full activation checkpointing. Local batch size 1. (Stats per GPU)

| Schedule | Sequence Length | Throughput (Tok/Sec) | Memory (GiB) |
|---|---|---|---|
| FSDP 8, CP 1 | 32,768 | 3,890 | 83.9 |
| FSDP 4, CP 2 | 65,536 | 2,540 | 84.2 |
| FSDP 2, CP 4 | 131,072 | 1,071 | 84.0 |
| FSDP 1, CP 8 | 262,144 | 548 | 84.5 |

Table 6: 4D parallelism (FSDP + TP + PP + CP) + `torch.compile` + Float8 + AsyncTP + 1F1B on Llama 3.1 405B model, 512 GPUs. Mixed precision training. Full activation checkpointing. TP degree 8, PP degree 8. Local batch size 8. (Stats per GPU)

| Schedule | Sequence Length | Throughput (Tok/Sec) | Memory (GiB) |
|---|---|---|---|
| FSDP 8, CP 1 | 32,768 | 76 | 75.3 |
| FSDP 4, CP 2 | 65,536 | 47 | 75.9 |
| FSDP 2, CP 4 | 131,072 | 31 | 77.1 |
| FSDP 1, CP 8 | 262,144 | 16 | 84.9 |

4D parallelism. This section highlights key observations and motivations for using TORCHTITAN 4D parallelism, focusing on a specific combination shown in Figure 2.

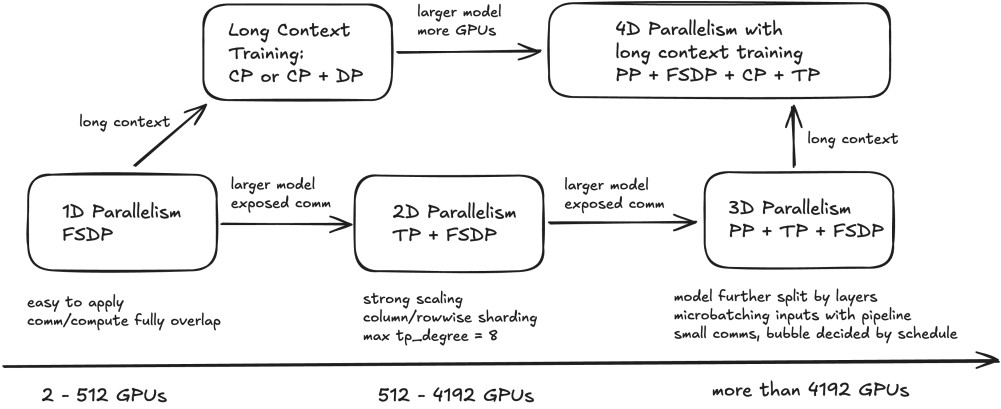

Figure 2: Scaling with 4D Parallelism

### 3.3.1 SCALING WITH FSDP

FSDP (ZeRO) is a general technique applicable to any model architecture and is often sufficient as the first degree of parallelism when communication is faster than computation (e.g., up to 512 GPUs). However, with larger scales, collective latency increases linearly with the world size, limiting efficiency. To overcome this, model parallelism like TP and PP can be combined with FSDP.

### 3.3.2 2D PARALLELISM: TP WITH FSDP

Tensor Parallelism (TP) reduces collective latency by distributing work across GPUs, enabling smaller effective batch sizes and reducing peak memory usage for large models or sequence lengths. Combining FSDP and TP allows strong scaling with a fixed problem/batch size (Details shown in Figure 4). TP also improves FLOP utilization by optimizing matrix multiplication shapes. However, TP introduces blocking collectives and is typically limited to intra-node scaling (e.g., NVLink), with degrees usually capped at 8. Scaling beyond 4192 GPUs requires combining TP with PP.

### 3.3.3 3D PARALLELISM: PP WITH 2D PARALLELISM

Pipeline Parallelism (PP) reduces communication bandwidth requirements by transmitting only activations and gradients between stages in a peer-to-peer manner. PP is particularly effective for mitigating FSDP communication latency at larger scales or in bandwidth-limited clusters. The efficiency of PP depends on pipeline schedules and microbatch sizes, which influence the size of pipeline "bubbles."

### 3.3.4 LONG CONTEXT TRAINING AND 4D PARALLELISM

Context Parallelism (CP) allows ultra long context training by splitting the context (sequence) dimension across GPUs to avoid OOM errors. CP is mainly used for long context training, to give the model capability to capture more correlations for tokens, thus enhancing the overall model quality. For scaling sequence length, CP can be used alone or together with DP. When training large models or on large number of GPUs, we can combine CP with 3D paralleism, where TP usually keeps the innner-most DeviceMesh dimension, and CP applies in the next outer DeviceMesh dimension.

## 4 RELATED WORK

Libraries such as Megatron-LM (Narayanan et al., 2021), DeepSpeed (Rasley et al., 2020), veScale (Inc., 2024) and PyTorch Distributed (Paszke et al., 2019; Meta Platforms, Inc., 2024) provide APIs for distributed workflows. However, these frameworks present challenges in flexibility, integration, and scalability. TORCHTITAN addresses these limitations with native support for key features absent in existing systems:

- *Megatron-LM*: Requires model modifications for TransformerEngine, lacks seamless FSDP integration with TP and PP, and does not support advanced pipeline schedules to minimize computation overhead.

- *DeepSpeed*: Depends on Megatron-LM for TP and CP, with limited support for FSDP and advanced pipeline schedules.

- *veScale*: Does not support FSDP, CP, SAC, Float8 training, or `torch.compile`, and offers only three pipeline schedules, compared to TORCHTITAN 's six.

We note that each of these libraries has its own strengths, and TORCHTITAN is designed to provide foundational components that can be leveraged by all of them. A detailed comparison, including feature breakdowns and code complexity analysis, is available in Appendix B.9. Slapo (Chen et al., 2023) introduces a schedule language to convert a PyTorch model for common model training optimizations such as 3D parallelism, and supports progressive optimization through high-level primitives. In contrast, TORCHTITAN provides modular and composable APIs built on DTensor and DeviceMesh.

## 5 CONCLUSION

TORCHTITAN is a powerful and flexible framework for LLM training, enabling seamless composability of parallelism techniques (FSDP, TP, PP, CP), memory optimizations (Float8, activation checkpointing), and PyTorch compiler integration for enhanced efficiency. Its modular design supports evolving architectures and hardware, fostering innovation with multi-axis metrics.

Designed for interpretability and production-grade training, TORCHTITAN offers elastic scalability, comprehensive training recipes, and expert guidance on distributed training strategies. As demonstrated in experiments, it accelerates training by 65.08% on Llama 3.1 8B (128 GPUs, 1D), 12.59% on Llama 3.1 70B (256 GPUs, 2D), and 30% on Llama 3.1 405B (512 GPUs, 3D) over optimized baselines, while enabling long-context training with 4D composability. With its robust features and high efficiency, TORCHTITAN is an ideal one-stop solution for challenging LLM training tasks.

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

## A  COMPOSABLE 4D PARALLELISM WALKTHROUGH

We have discussed the scaling with TORCHTITAN 4D parallelism and the motivations to apply different parallelisms to scale training to thousands of GPUs. In this section we will walk through the 4D parallelism code in TORCHTITAN.

The first step is to create an instance of the model (e.g. the `Transformer` for Llama models) on the meta device. We then apply PP by splitting the model into multiple PP stages according to the `pipeline_parallel_split_points` config. Note that for PP with looped schedules, we may obtain multiple `model_parts` from PP splitting, where each item in `model_parts` is

one stage-model-chunk. Next we apply SPMD-style distributed training techniques including TP, activation checkpointing, torch.compile, FSDP, and mixed precision training for each model part, before actually initializing the sharded model on GPU.

```python
# meta init
with torch.device("meta"):
   model = model_cls.from_model_args(model_config)

# apply PP
pp_schedule, model_parts = models_pipelining_fns[model_name](
   model, pp_mesh, parallel_dims, job_config, device, model_config,
      loss_fn
)

for m in model_parts:
   # apply SPMD-style distributed training techniques
   models_parallelize_fns[model_name](m, world_mesh, parallel_dims,
      job_config)
   # move sharded model to GPU and initialize weights via DTensor
   m.to_empty(device="cuda")
   m.init_weights()
```

To apply PP to the model, we run the following code at the high level. `pipeline_llama_manual_split` splits the model into multiple stages according to the manually given `pipeline_parallel_split_points` config, by removing the unused model components from a complete model (on the meta device). Then `build_pipeline_schedule` make the pipeline schedule with various options from `torch.distributed.pipelining`, including 1F1B (Narayanan et al., 2019), GPipe (Huang et al., 2019), interleaved 1F1B (Narayanan et al., 2021), etc. instructed by the `pipeline_parallel_schedule` config.

```python
stages, models = pipeline_llama_manual_split(
   model, pp_mesh, parallel_dims, job_config, device, model_config
)
pp_schedule = build_pipeline_schedule(job_config, stages, loss_fn)
return pp_schedule, models
```

TP and FSDP are applied in the SPMD-style `models_parallelize_fns` function. To apply TP, we utilize the DTensor `parallelize_module` API, by providing a TP "plan" as the instruction of how model parameters should be sharded. In the example below, we showcase the (incomplete) code for sharding the repeated `TransformerBlock`.

```python
for layer_id, transformer_block in model.layers.items():
   layer_tp_plan = {
      "attention_norm": SequenceParallel(),
      "attention": PrepareModuleInput(
         input_layouts=(Shard(1), None),
         desired_input_layouts=(Replicate(), None),
      ),
      "attention.wq": ColwiseParallel(),
      ...
   }
   parallelize_module(
      module=transformer_block,
      device_mesh=tp_mesh,
      parallelize_plan=layer_tp_plan,
   )
```

Then, we apply the FSDP by wrapping each individual `TransformerBlock` and then the whole model. Note that the FSDP2 implementation in PyTorch comes with mixed precision training support. By default, we use `torch.bfloat16` on parameters all-gather and activation computations, and use `torch.float32` on gradient reduce-scatter communication and optimizer updates.

```python
mp_policy = MixedPrecisionPolicy(param_dtype, reduce_dtype)
fsdp_config = {"mesh": dp_mesh, "mp_policy": mp_policy}

for layer_id, transformer_block in model.layers.items():
    # As an optimization, do not reshard_after_forward for the last
    # TransformerBlock since FSDP would prefetch it immediately
    reshard_after_forward = int(layer_id) < len(model.layers) - 1
    fully_shard(
        transformer_block,
        **fsdp_config,
        reshard_after_forward=reshard_after_forward,
    )
fully_shard(model, **fsdp_config)
```

Independently, we can apply CP by running each training iteration under a Python context manager.

```python
optional_context_parallel_ctx = (
    utils.create_context_parallel_ctx(
        cp_mesh=world_mesh["cp"],
        cp_buffers=[input_ids, labels] + [m.freqs_cis for m in
            model_parts],
        cp_seq_dims=[1, 1] + [0 for _ in model_parts],
        cp_no_restore_buffers={input_ids, labels},
        cp_rotate_method=job_config.experimental.context_parallel_rotate_method,
    )
    if parallel_dims.cp_enabled
    else None
)
...
with train_context(optional_context_parallel_ctx):
    pred = model(input_ids)
    loss = loss_fn(pred, labels)
```

# B    SUPPLEMENTARY MATERIALS

## B.1    FULLY SHARDED DATA PARALLEL

FSDP2 advances the tensor sharding approach by replacing the original FSDP1 FlatParameter sharding. Specifically, parameters are now represented as DTensors sharded on the tensor dimension 0. This provides better composability with model parallelism techniques and other features that requires the manipulation of individual parameters, allowing sharded state dict to be represented by DTensor without any communication, and provides for a simpler meta-device initialization flow via DTensor. For example, FSDP2 unlocks finer grained tensor level quantization, especially Float8 tensor quantization, which we will showcase in the results section.

As part of the rewrite from FSDP1 to FSDP2, FSDP2 implements an improved memory management system by avoiding using record stream. This enables deterministic memory release, and as a result provides lower memory requirements per GPU relative to FSDP1. For example on Llama 2 7B, FSDP2 records an average of 7% lower GPU memory versus FSDP1.

In addition, by writing efficient kernels to perform multi-tensor allgather and reduce scatter, FSDP2 shows on-par performance compared to FSDP1, with even slight performance gains - using the Llama 2 7B, FSDP2 shows an average gain of 1.5% faster throughput.

The performance gains are the result of employing two small performance improvements. First, only a single division kernel is run for the FP32 reduce scatter (pre-dividing the local FP32 reduce-scatter gradient by world size, instead of a two step pre and post divide by square root of world size). Secondly, in TORCHTITAN, FSDP2 is integrated with a default of not re-sharding the final block in a transformer layer during the forward pass, since it will be immediately re-gathered at the start of the backward pass.

**Usage**: TORCHTITAN has fully integrated FSDP2 as the default parallelism when training, and the `data_parallel_shard_degree` is the controlling dimension in the command line or TOML file. Note that for ease of use, the default `data_parallel_shard_degree` is -1, means to simply use all GPUs available, so user do not need to specify the actual world size.

## B.2 HYBRID SHARDED DATA PARALLEL

Hybrid Sharded Data Parallel (HSDP) is an extension of FSDP (Zhang et al., 2022). In FSDP, communication occurs between all devices within the FSDP group. However, at some point, the FSDP communication overhead exceeds its corresponding computation because the latency of allgather/reduce-scatter communications increases linearly with the number of devices. This results in low MFU and becomes worthless to add more GPUs for scaling.

HSDP obviates this to some degree by creating a 2-D DeviceMesh that contains replica groups on one dimension and shard groups on the other dimension, where each shard group runs FSDP and the replica group runs normal data parallel. This ensures the FSDP communications happen in a fraction of the original world size, with the addition of backward gradient allreduce across replica groups. HSDP reduces FSDP communication overhead and allows further scaling with data parallel.

**Usage**: TORCHTITAN makes it easy to experiment with HSDP by using the two configurable settings: `data_parallel_shard_degree` and `data_parallel_replicate_degree`, which controls the degree of the shard and replica groups we are creating. The product of both replicate and shard degree is the actual data parallel world size.

## B.3 TENSOR PARALLEL

TP partitions the attention and feed forward network (MLP) modules of a transformer layer across multiple devices, where the number of devices used is the TP degree. This allows for multiple GPUs to cooperatively process the same batch by using the local sharded model parameters, at the cost of adding `all-reduce/all-gather/reduce-scatter` operations to synchronize intermediate activations.

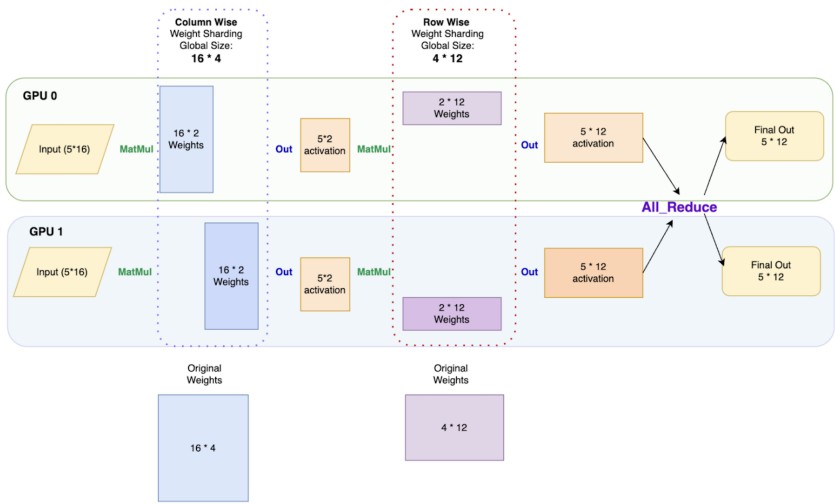

Figure 3: Tensor Parallel in detail (2 GPUs, data moves from left to right).

Due to the additional collectives introduced by TP, it needs to happen within a fast network (i.e NVLink). When training LLMs, TP is usually combined with FSDP, where TP shards within nodes and FSDP shards across nodes to create the 2D hierarchical sharding on different DeviceMesh dimensions.

**Usage**: Because of the synergistic relationship between TP and SP, TORCHTITAN natively bundles these two together and they are jointly controlled by the TP degree setting in the command line

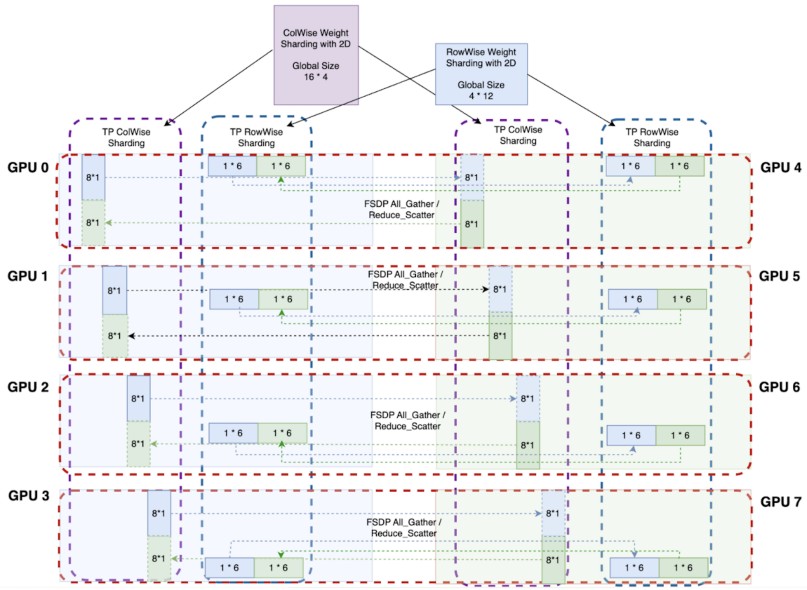

Figure 4: FSDP2 + Tensor Parallel (TP degree 4) sharding layout, with 2 nodes of 4 GPUs.

or the TOML entry of `tensor_parallel_degree`. Setting this to 2 for example would mean that 2 GPUs within the node will share the computational load for each transformer layers attention and MLP modules via TP, and normalization/dropout layers via Sequence Parallel. Loss Parallel is implemented via a context manager as it needs to control the loss computation outside of the model's forward computation. It can be enabled via `enable_loss_parallel`.

## B.4    PIPELINE PARALLEL

We expose several parameters to configure PP. `pipeline_parallel_degree` controls the number of ranks participating in PP. `pipeline_parallel_split_points` accepts a list of strings, representing layer fully-qualified-names before which a split will be performed. Thus, the total number of pipeline stages $V$ will be determined by the length of this list. `pipeline_parallel_schedule` accepts the name of the schedule to be used. If the schedule is multi-stage, there should be $V > 1$ stages assigned to each pipeline rank, otherwise $V == 1$. `pipeline_parallel_microbatches` controls the number of microbatches to split a data batch into.

## B.5    ENABLING 4D PARALLEL TRAINING: CONTEXT-PARALLEL (CP)

To address context scaling, we have incorporated Context Parallelism (CP) into TORCHTI-TAN. Following the principles of modular design of TORCHTITAN, CP was integrated via a context manager that dynamically replaces calls to attention operators (namely, `scaled_dot_product_attention`) with CP operations, ensuring no changes to the model code are required.

Under the hood, CP shards the DTensor along the sequence dimension across the CP device mesh. It extends the DTensor dispatcher to handle CP-specific operations, such as Ring Attention and causal attention load balancing, ensuring efficient operation. By extending DTensor's capabilities to support CP, TORCHTITAN ensures that CP is fully compatible with all other parallelisms (FSDP, TP, PP), optimizations (e.g., activation checkpointing, `torch.compile`), and DCP. This demonstrates the extensibility of TORCHTITAN 's modular design, which accommodates future optimizations seamlessly while maintaining performance and compatibility.

### B.6 ACTIVATION CHECKPOINTING

TORCHTITAN offers two types of Selective Activation Checkpointing which allow for a more nuanced tradeoff between memory and recomputation. Specifically, we offer the option to selectively checkpoint "per layer" or "per operation". The goal for per operation is to free memory used by operations that are faster to recompute and save intermediates (memory) for operations that are slower to recompute and thus deliver a more effective throughput/memory trade-off.

**Usage:** AC is enabled via a two-line setting in the command line or TOML file. Specifically, `mode` can be either `none`, `selective`, or `full`. When `selective` is set, then the next config of `selective_ac_type` is used which can be either a positive integer to enable selective layer checkpointing, or `op` to enable selective operation checkpointing. Per layer takes an integer input to guide the checkpointing policy, where 1 = checkpoint every layer (same as full), 2 = checkpoint every other layer, 3 = checkpoint every third layer, etc. Per op(eration) is driven by the `_save_list` policy in `parallelize_llama.py` which flags high arithmetic intensity operations such as matmul (matrix multiplication) and SPDA (Scaled Dot Product Attention) for saving the intermediate results, while allowing other lower intensity operations to be recomputed. Note that for balancing total throughput, only every other matmul is flagged for saving.

### B.7 ASYNCTP

The `SymmetricMemory` collectives used in AsyncTP are faster than standard NCCL collectives and operate by having each GPU allocate an identical memory buffer in order to provide direct P2P access. `SymmetricMemory` relies on having NVSwitch within the node, and is thus generally only available for H100 or newer GPUs.

**Usage**: AsyncTP is enabled within the experimental section of the TORCHTITAN TOML config file and turned on or off via the `enable_async_tensor_parallel` boolean setting.

### B.8 CUSTOMIZING FSDP2 MIXED PRECISION IN TORCHTITAN

Mixed Precision is controlled by the `MixedPrecisionPolicy` class in the `apply_fsdp` function, which is then customized with `param_dtype` as BF16, and `reduce_dtype` defaulting to FP32 by default in TORCHTITAN. The `reduce_dtype` in FP32 means that the reduce-scatter in the backwards pass for gradient computation will take place in FP32 to help maximize both stability and precision of the gradient updates.

### B.9 TORCHTITAN: COMPREHENSIVE FEATURE SET AND REDUCED COMPLEXITY

#### B.9.1 TORCHTITAN ENABLES NEW DESIGNS

TORCHTITAN 's extensive feature set and broad design space coverage are driven by its unified design principles i.e. modularity, composability, and extensibility. Leveraging these principles, TORCHTITAN seamlessly integrates diverse parallelism strategies (FSDP, TP, PP, and CP) and optimizations (e.g., SAC, Float8 training). This unified framework not only supports advanced pipeline schedules and multi-dimensional parallelism but also simplifies the integration of new techniques, making it highly adaptable for cutting-edge research and production-grade deployments.

The following table highlights TORCHTITAN 's capabilities in context of parallelism, checkpointing and compiler support offerings compared to Megatron-LM, DeepSpeed, and veScale:

#### B.9.2 CODE COMPLEXITY AND MAINTAINABILITY

TORCHTITAN 's design principles also contribute to its significantly reduced code complexity. Despite offering a rich feature set, TORCHTITAN maintains a compact and modular codebase, making it easier to extend, maintain, and evolve while ensuring high performance. The following table compares the lines of code (LOC) for TORCHTITAN with Megatron-LM and DeepSpeed:

---

[4]Custom Fusion Kernels

Table 7: Comparison of TORCHTITAN with Megatron-LM, DeepSpeed, and veScale with respect to parallelism, compiler support, activation checkpointing, and model checkpointing.

| Features | TORCHTITAN | Megatron-LM | DeepSpeed | veScale |
|---|---|---|---|---|
| FSDP-Zero2 | Yes | Yes | Yes | No |
| FSDP-Zero3 | Yes | Yes | Yes | No |
| HSDP | Yes | Yes | No | No |
| TP | Yes | Yes | No | Yes |
| Async TP (Micro-pipelining) | Yes | Yes | No | Yes |
| CP | Yes | Yes | No | No |
| PP-Gpipe | Yes | Yes | Yes | No |
| PP-Interleaved (1F1B) | Yes | Yes | Yes | Yes |
| PP-Looped-BFS | Yes | No | No | No |
| PP-1F1B | Yes | Yes | Yes | Yes |
| PP-Flexible-Interleaved-1F1B | Yes | No | No | No |
| PP-ZeroBubble | Yes | No | No | Yes |
| (TP+SP)+PP | Yes | Yes | No | Yes |
| DDP+(TP+SP)+PP | Yes | Yes | No | Yes |
| FSDP+(TP+SP) | Yes | No | No | No |
| FSDP+(TP+SP)+PP | Yes | No | No | No |
| FSDP+(TP+SP)+PP+CP | Yes | No | No | No |
| MoE | Ongoing | Yes | No | No |
| Full AC | Yes | Yes | Yes | Yes |
| Flexible SAC | Yes | No | No | No |
| DCP | Yes | Yes | Yes | Yes |
| Float8 Training | Yes | Yes | No | No |
| `torch.compile` | Yes | No[4] | Partial | No |

Table 8: Lines of Code (LOC) comparison across systems.

| Lines of Code (LOC) | TORCHTITAN | Megatron-LM | DeepSpeed |
|---|---|---|---|
| Core Codebase | 7K | 93K | 94K |
| Total Codebase (Including Utils) | 9K | 269K | 194K |

## B.10 EXTENDED EXPERIMENTS ANALYSIS: PERFORMANCE AND LOSS CONVERGING

### B.10.1 PERFORMANCE

Our experiments in Section 3.2 serve multiple objectives:

- **Establish composability and modularity:** TORCHTITAN demonstrates seamless integration of various parallelisms and optimization techniques.
- **Showcase performance improvements:** Significant speed-ups are observed across parallelisms and optimizations.
- **Validate elastic scalability:** TORCHTITAN scales effectively with both the model size and the number of GPUs.
- **Ablation studies:** Detailed performance gains for individual techniques are presented.

In particular

- Table 1: Highlights improvements from compiler support over eager execution, followed by further gains with Float8 training.
- Table 2: Demonstrates how earlier gains scale as the number of GPUs increases.
- Table 3: Shows speed-up achieved by AsyncTP (a HW/SW co-designed technique) over 2D training combined with `torch.compile` and Float8 training.

- Table 4: Quantifies the benefits of Interleaved 1F1B scheduling over 1F1B on top of AsyncTP, `torch.compile`, and Float8 training.

- Table 5: Demonstrates the effectiveness of CP on enabling long context training, even at small scale.

- Table 6: Demonstrate the composability of 4D parallelism, and the effectiveness of CP on enabling long context training at large scale.

For FSDP, the ZeRO-3 variant is used for all experiments except for those involving PP where the ZeRO-2 variant is used. This distinction is due to the inefficiency of ZeRO-3 in PP, where it incurs additional all-gather calls for each microbatch. In contrast, ZeRO-2 gathers parameters only once for the first microbatch and reshards after the last microbatch's backward pass.

### B.10.2 LOSS CONVERGING

TORCHTITAN 's design principles have influenced the development of advanced distributed training features such as FSDP2, AsyncTP, PP, and CP in PyTorch's distributed library. Throughout these contributions, we have ensured the loss converging of individual techniques as well as their various combinations of parallelisms and optimizations.

For example, below is a series of loss-converging tests covering both parallelisms and training optimizations. We use notations of "FSDP 8" for an experiment in which the degree of FSDP is 8, "FSDP 8, CP 8" for an experiment on 64 GPUs where FSDP degree is 8 and CP degree is 8, etc. We assume the correctness of FSDP, which can be further verified by comparing it with DDP or even single-device jobs.

Table 9: Loss-converging tests setup.

| Parallelism | Techniques |
| --- | --- |
| FSDP 8 (ground truth) | default |
| FSDP 8, TP 2, PP 2 | torch.compile, Float8, async TP, Interleaved 1F1B |
| FSDP 8, TP 2, CP 2, PP 2 | torch.compile, Float8, async TP, Interleaved 1F1B |
| FSDP 8, CP 8 | default |

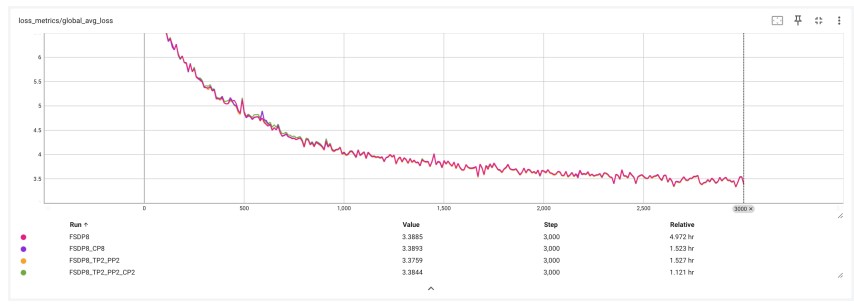

Figure 5: Loss converging tests on Llama 3.1 8B. C4 dataset. Local batch size 4, global batch size 32. 3000 steps, 600 warmup steps.

