# OpenReview forum: "TorchTitan: One-stop PyTorch native solution for production ready LLM pretraining"
_ICLR.cc/2025/Conference — ICLR 2025 Poster_

### Official Review · Reviewer_UCk4 · 2024-11-02

**Soundness:** 4
**Presentation:** 3
**Contribution:** 4
**Rating:** 10
**Confidence:** 5

**Summary:**

This paper introduces LEGOSCALE, an open-source, PyTorch-native distributed training system that unifies and advances state-of-the-art techniques, streamlining integration and reducing engineering overhead. LEGOSCALE enables 3D parallelism in a modular and composable featuring elastic scaling to adapt to changing computational requirements. By providing an accessible and extensible platform, LEGOSCALE democratizes large language model pre-training, empowering a wider range of researchers and developers to tap into the potential of LLMs and accelerate innovation in the field.

**Strengths:**

Thank you for submitting this paper to ICLR! I loved reading this paper.

1. Important problem. LLM training is challenging and resource-consuming.

2. Very graceful, interesting and timely solution. Pytorch-native is what the community needs, which is different with existing systems like Megatron.

**Weaknesses:**

1. Too many modules are introduced in limited pages, which is not friendly to general readers without machine learning system background.

2. Limited model support. I would like to see more application on MoE model, Multi-modal LLM, etc.

3. 3D-parallelism is not comprehensive / state-of-the-art at the time of 2024. Consider adding more support on context parallel (also mentioned in paper), expert parallel, etc.

**Questions:**

Although this work is more of an engineer's contribution than a research novelty contribution, I still think it is important and should be accepted by ICLR. There are some questions I would like to see more discussion of.

1. For Pytorch native training framework, there is another framework implemented by Bytedance: https://github.com/volcengine/veScale. Could you compare the differences between them? It seems they support more parallisms.

2. I would like to see some results of end-to-end throughput comparison with the latest Megatron, Deepspeed.

---

> ### Author Response · Authors · 2024-11-21
>
> We are grateful to the reviewer for acknowledging, appreciating, validating, and providing critical feedback on our work. Please consider our response labeled as Wi, which aims to address weaknesses i, while Qi aims to address Question i, pointed out by the reviewer.
>
>
> **W1:**
>
> We agree with the reviewer that presenting a holistic and comprehensive overview of LEGOSCALE may feel overwhelming to readers without a background in machine learning systems. However, we believe it is crucial to provide a complete picture of the pre-training system’s capabilities, including its parallelisms and memory and compute optimization techniques. To address this, we have kept the descriptions at a high level, focusing on the functionality of each module while avoiding excessive implementation details.
>
> We will carefully review the paper and make adjustments to ensure the right balance between accessibility for general readers and sufficient depth for those with ML systems expertise. This will help make LEGOSCALE’s design and contributions clear to a broader audience while maintaining technical rigor.
>
> **W2:**
>
> This is a great suggestion that will highlight LEGOSCALE’s model-agnostic and general design. While demonstrated with the Llama model, **LEGOSCALE is not model-specific**. Its techniques are general-purpose, allowing seamless integration of other models without changes to the system.
>
> We plan to release **Llama 3.2 multimodal recipes** soon and are working on integrating diffusion models, showcasing LEGOSCALE’s flexibility and extensibility.
>
> **W3:**
>
> We thank the reviewer for highlighting the importance of context scaling and its integration into LEGOSCALE. We completely agree with the reviewer regarding its significance and are pleased to report that, since submitting the paper, we have successfully incorporated **Context Parallelism (CP)** into LEGOSCALE. We will include details of this in the camera-ready version of the paper. Below, we provide an outline of how LEGOSCALE addresses context scaling through CP.
>
> ### Context Parallelism (CP) in LEGOSCALE
> To address **context scaling**, LEGOSCALE extends DTensor’s capabilities to support CP. Following LEGOSCALE’s modular design principles, CP was integrated using two user-friendly APIs:
> 1. **Module Wrapper**: Similar to TP, for distributing the Attention module.
> 2. **Context Manager**: Dynamically replaces `scaled_dot_product_attention` calls with context-parallel operators, ensuring no changes to the model code.
>
> Under the hood, CP:
> - Shards the **DTensor** along the sequence dimension across the CP device mesh.
> - Extends the DTensor dispatcher to handle **context-parallel-specific operations**, such as **ring-attention** and **causal attention load balancing**, ensuring efficient operation.
>
> By extending DTensor’s design, CP in LEGOSCALE remains fully compatible with other parallelisms (FSDP, TP, PP), optimizations (e.g., activation checkpointing, `torch.compile`), and **Distributed Checkpointing (DCP)**. This demonstrates LEGOSCALE’s extensibility and future-proof design, accommodating new optimizations seamlessly while maintaining performance and compatibility.
>
> ### Context Scaling Results (Llama3-8B, 64 H100 GPUs, TP=8, DPxCP=8)
>
> | **CP Degree** (GPUs on CP dimension) | **Max Context Length** (Without OOM) | **Local WPS** (linear scaling) | **MFU** (slight degradation) |
> |---------------------------------------|---------------------------------------|--------------------------------|-------------------------------|
> | 1                                     | 32,768                                | 4497                           | 43.09                         |
> | 2                                     | 81,920                                | 1970                           | 34.63                         |
> | 4                                     | 147,456                               | 1205                           | 33.73                         |
> | 8                                     | 294,912                               | 638                            | 34.09                         |
>
> The integration of Context Parallelism highlights LEGOSCALE’s ability to adapt to emerging requirements in LLM training. Its unified DTensor abstraction and modular framework ensure that context scaling and other advancements can be seamlessly incorporated, making it a robust solution for future research and deployment.
>
> The efforts for adding expert parallelism to the framework is underway.

---

> > ### Author Response · Authors · 2024-11-21
> >
> > **Q1 and Q2:**
> >
> > We thank the reviewer for their comments and will include a detailed discussion on related work in the paper to contextualize LEGOSCALE’s contributions relative to DeepSpeed, Megatron-LM, and veScale.
> >
> > LEGOSCALE’s unified design principles—modularity, composability, and extensibility—enable seamless integration of FSDP, TP, PP, CP, and optimizations like SAC and Float8 training. This framework supports advanced pipeline schedules, multi-dimensional parallelism, and simplifies integration of new techniques for research and production.
> >
> > **Feature Comparison**
> >
> > LEGOSCALE addresses key limitations:
> >
> > Megatron-LM: Requires significant model modifications and lacks advanced pipeline schedules and flexibility in combining FSDP with TP/PP.
> >
> > DeepSpeed: Relies on Megatron-LM for TP/CP and lacks integration of advanced schedules with FSDP.
> >
> > veScale: Does not support FSDP, CP, SAC, Float8 training, or full torch.compile compatibility. It offers three pipeline schedules compared to LEGOSCALE’s six and treats SP as a separate dimension, implementing 3D instead of 4D parallelism.
> >
> > The following table highlights LEGOSCALE’s broader capabilities.
> >
> > | **Features**                          | **Legoscale** | **Megatron-LM**            | **DeepSpeed** | **VeScale** |
> > |---------------------------------------|---------------|-----------------------------|---------------|-------------|
> > | FSDP-Zero2                            | Y             | Y                           | Y             | N           |
> > | FSDP-Zero3                            | Y             | Y                           | Y             | N           |
> > | HSDP                                  | Y             | Y                           | N             | N           |
> > | TP                                    | Y             | Y                           | N             | Y           |
> > | Async TP (Micro-pipelining)           | Y             | Y                           | N             | Y           |
> > | CP                                    | Y             | Y                           | N             | N           |
> > | PP-Gpipe                              | Y             | Y                           | Y             | N           |
> > | PP-Interleaved (1F1B)                 | Y             | Y                           | Y             | Y           |
> > | PP-Looped-BFS                         | Y             | N                           | N             | N           |
> > | PP-1F1B                               | Y             | Y                           | Y             | Y           |
> > | PP-Flexible-Interleaved-1F1B          | Y             | N                           | N             | N           |
> > | PP-ZeroBubble                         | Y             | N                           | N             | Y           |
> > | (TP+SP)+PP                            | Y             | Y                           | N             | Y           |
> > | DDP+(TP+SP)+PP                        | Y             | Y                           | N             | Y           |
> > | FSDP +(TP + SP)                       | Y             | N                           | N             | N           |
> > | FSDP + (TP + SP) + PP                 | Y             | N                           | N             | N           |
> > | FSDP + (TP + SP) + PP + CP            | Y             | N                           | N             | N           |
> > | MoE                                   | Ongoing       | Y                           | N             | N           |
> > | Full AC                               | Y             | Y                           | Y             | Y           |
> > | Flexible SAC                          | Y             | N                           | N             | N           |
> > | DCP                                   | Y             | Y                           | Y             | Y           |
> > | Float-8 Training                      | Y             | Y                           | N             | N           |
> > | torch.compile                         | Y             | N (Custom Fusion Kernels)   | Partial       | N           |
> >
> > **Code Complexity:**
> >
> > These same design principles also contribute to LEGOSCALE’s reduced code complexity. Despite offering a rich feature set, LEGOSCALE has a compact and maintainable codebase, significantly smaller than other systems. Its modular architecture ensures extensibility and performance while minimizing complexity, making the framework easier to maintain and evolve.
> >
> > | **Lines of Code (LOC)**             | **Legoscale** | **Megatron-LM** | **DeepSpeed** |
> > |-------------------------------------|---------------|------------------|---------------|
> > | **Core Codebase**                   | 7K            | 93K             | 94K           |
> > | **Total Codebase (Including Utils)**| 9K            | 269K            | 194K          |
> >
> > LEGOSCALE achieves comparable or superior functionality with a fraction of the codebase, making it easier to maintain, extend, and integrate.

---

> > > ### Comment · Reviewer_UCk4 · 2024-12-01
> > >
> > > Thanks for answering my questions. It's a good paper and I raised my score. I would like to champion this work for acceptance by ICLR.

---

> > > > ### Author Response · Authors · 2024-12-02
> > > >
> > > > Thank you for your kind words and for raising your score. We greatly appreciate your support in championing this work for acceptance at ICLR. We are excited about the opportunity to share LEGOSCALE with the research community.

---

### Official Review · Reviewer_DprR · 2024-11-03

**Soundness:** 3
**Presentation:** 3
**Contribution:** 2
**Rating:** 6
**Confidence:** 3

**Summary:**

This submission comprises a technical report / white paper about the development of LegoScale, a pytorch-based solution for scalable LLM training in distributed settings. The proposed framework extends the capabilities of vanilla pytorch for LLM training in many aspects, ranging from improved scalability through a principled combination of numerous optimised parallelism techniques and memory management, to improved logging and debugging mechanisms.

**Strengths:**

- The proposed framework offers a streamlined approach for LLM training across distributed systems with different scales of compute. This scalability is achieved through a principled and well-engineered framework that allows tunable exploitation of different parallelism dimensions and other optimisations. This abstracts tons of complexities from deep learning practitioners.
- Evidently, the majority of design choices in the proposed framework have relied on insights and experience from both the LLM training and the GPU deployment communities.
- The reported experiments demonstrate that the proposed framework can achieve notable speed-ups at training time.

**Weaknesses:**

- It is difficult to identify much research novelty in the proposed framework, since its contribution can mostly be found in the design and tons of non-trivial engineering efforts behind LegoScale.
- It is unclear whether this contribution will last or may wind down due to enhancements of the original pytorch framework towards the same direction, or lack of maintenance on the proposed framework which may lead to lack of support from latest pytorch versions or for latest GPU models.

**Questions:**

This review is intentionally short and high-level, due to the technical excellence, but also lack of research novelty, of the proposed system.
In my opinion, white papers of successful training frameworks can still be published in ML venues; but will be willing to adjust my score considering the outcome of the upcoming discussion with other reviewers'.

A piece of information that can facilitate this process is any insights that the authors have obtained and can share with us about the usage of their framework from the community (without violating the double-blind process). Has the framework been open-sourced? How many people are actively contributing to it? How many people are currently using it? Are the reported results easily replicated out-of-the-box on different clusters? Is long-term support and maintenance of the framework part  of the plans of the authors?

Post rebuttal edit:
 I am increasing my score from 5 to 6. Although I maintain the position that the novelty in the manuscript is limited, I believe that the proposed framework can have notable impact to the community, and as such it is worth being presented to an ML venue in my opinion.

---

> ### Author Response · Authors · 2024-11-21
>
> We thank the reviewer for underscoring the importance of our work, recognizing its performance, appreciating our design, and providing detailed and critical feedback on our work. Please consider our response labeled as Wi, which aims to address weaknesses i, while Qi aims to address Question i, pointed out by the reviewer.
>
> **W1 and Q1:**
>
> LEGOSCALE’s key research contribution lies in analyzing and identifying a common set of design principles across parallelism and optimization techniques. These principles unify diverse methods into a cohesive system. The engineering contribution is in enforcing these principles across individual techniques, resulting in a modular, composable, and flexible distributed system that enables seamless scaling, efficient deployment, and integration of new techniques. These contributions have shaped PyTorch features like FSDP2, Async-TP, PP, and DTensor advancements.
>
> **LEGOSCALE’s Design Principles**
>
> LEGOSCALE adopts a top-down approach, enforcing modularity, non-intrusiveness, and extensibility. By leveraging DTensor and DeviceMesh, it provides a unified framework for diverse parallelisms and optimizations, ensuring scalability and compatibility with PyTorch.
>
> **Advancements to DTensor**
>
> DTensor sharding was extended to support n-D parallelism, integrating TP, CP, and PP. Compatibility with torch.compile was added for compiler optimizations, and checkpointing efficiency improved with robust state dict support. Production readiness was bolstered by resolving critical bugs.
>
> **Contribution to PyTorch FSDP2**
>
> FSDP2 replaced FlatParameter with DTensors sharded on dim-0, enabling flexible parameter handling (e.g., freezing, fp8 all-gather). It integrates with torch.compile, reduces GPU memory usage by eliminating CPU synchronization, and simplifies checkpointing with sharded state dicts.
>
> **Contribution to PyTorch Async-TP/CP**
>
> Async-TP introduced fused operators like fused_all_gather_matmul to reduce synchronization overheads, enabling fused compute and communication. This design facilitates new parallelisms like CP and minimizes bottlenecks with CUDA streams and SymmetricMemory.
>
> **Contribution to PyTorch PP**
>
> Pipeline Parallelism incorporates LEGOSCALE’s principles, enabling dynamic schedules, fine-grained stage control, and the addition of new schedules while integrating with TP and DP.
>
> **A Unified Template for Distributed Training**
>
> LEGOSCALE provides a unified design template, simplifying exploration of configurations and enabling rigorous evaluation of techniques. It unifies, modularizes, and extends distributed methods, shaping PyTorch’s library and advancing scalable, flexible systems.
>
> **W2:**
>
> We have been actively involved with PyTorch developers and have significantly contributed to the design and enhancement of the PyTorch distributed library, as discussed earlier. By making LEGOSCALE fully PyTorch-native, we ensure seamless compatibility with upstream PyTorch developments, enabling users to benefit from the latest features and support for new GPU models. Despite offering a full range of 4D parallel training capabilities and optimizations, LEGOSCALE’s codebase is only 9K lines, making it highly maintainable and adaptable for future advancements.
>
> **Q2:**
>
> Thank you for the question. We have fully open-sourced the LEGOSCALE system on GitHub (anonymized to comply with ICLR's double-anonymous requirements). The framework has garnered significant interest, with over **2,600 stars**, **204 forks**, **469 closed pull requests**, and **35 active contributors**. LEGOSCALE enables researchers and practitioners to explore, develop, and scale advanced parallelism and optimization techniques in a production-grade pre-training system.
>
> LEGOSCALE has been validated on multiple platforms, including the public cloud (AWS), on-premises clusters of a Fortune 500 company, a leading private organization, and a world-renowned university. Results are designed to be easily replicable across various cluster configurations. The convergence graphs can be found [here](https://www.dropbox.com/scl/fo/w7pdixb5a8m8kyffycsxe/AMYZPLKvMutEfgPQSSiyDYE?rlkey=qm8i648i95b4s52eya722tej7&e=1&st=43aa066z&dl=0).
>
> We are committed to actively maintaining and supporting LEGOSCALE for the foreseeable future.
>
> Since submitting the paper, we have incorporated **Context Parallelism (CP)** and added three additional **Pipeline Parallelism (PP)** schedules (Looped-BFS, Flexible-Interleaved-1F1B, ZeroBubble).
>
> Another key feature is LEGOSCALE’s model-agnostic and general design. While demonstrated with the Llama model, **LEGOSCALE is not model-specific**. Its techniques are general-purpose, allowing seamless integration of other models without changes to the system.
>
> Our goal in submitting this paper is to raise awareness within the research community about a system that empowers advanced research on pre-training large language models (LLMs) at scale, fostering innovation in this critical area.

---

> > ### Comment · Reviewer_DprR · 2024-12-02
> >
> > Thank you for the detailed response. I have no further questions.
> >
> > Having carefully read the discussion so far, I believe that although the novelty of this submission remains rather limited, the underlying tool can have significant impact and contribution to the community. As such I am inclined to increase my score.

---

> > > ### Author Response · Authors · 2024-12-03
> > >
> > > Thank you for updating your score and recognizing LEGOSCALE’s potential impact on the community.

---

### Official Review · Reviewer_5X5L · 2024-11-03

**Soundness:** 4
**Presentation:** 3
**Contribution:** 3
**Rating:** 6
**Confidence:** 4

**Summary:**

This paper describes a feature-rich PyTorch native hardware-software system for efficiently training of LLMs.

**Strengths:**

The paper describes a useful tool that addresses an important problem.

The tool substantially improves training time.

The system appears to have sufficient features for production use.

The paper has some tutorial value in that it briefly explains a wide range of features appropriate for pre-training systems.

I think this paper's greatest value is in advertizing the pre-training system to researchers and developers who may want to use it.

**Weaknesses:**

There are few new ideas. The focus is on  making a powerful tool by building on existing ideas. However, this weakness is partially offset by the fact that the work may enable others to develop new ideas by reducing the time required for pre-training.

Interest will be bimodal: those who need to pre-train LLMs will be very interested, especially if they have access to the pre-training system, and those who do not will probably find little of interest in the paper.

There is not a strong distinction between engineering efforts to add features to the pre-training system and research efforts to develop new system design ideas. The paper explains the design of the tool but does little to justify design decisions. Improvements to writing, alone, might address these problem.

With the exception of the sensitivity analysis in the tables on page 8, the paper does not contain scientific findings.

Figures 1 and 2 are essentially impossible to read without magnification, which circumvents conference page limits.

**Questions:**

Is there a plan to make the software components of the pre-training system available to other researchers?

What do you view as the most novel new ideas within the design of your pre-training system?

---

> ### Author Response · Authors · 2024-11-21
>
> We are grateful to the reviewer for acknowledging the importance of our work, recognizing its performance, appreciating its comprehensiveness, and providing detailed and critical feedback on it. Please consider our response labeled as Wi, which aims to address weaknesses i, while Qi aims to address Question i, pointed out by the reviewer.
>
> **W1**:
>
> We thank the reviewer for highlighting that LEGOSCALE enables machine learning practitioners and researchers to focus on mathematical problems, architectural exploration, and training dynamics by abstracting away the complexities of distributed systems. Simultaneously, LEGOSCALE provides ML systems researchers with a unified, modular, composable, and extensible testbed, empowering them to invent, extend, and swap newly developed techniques across various parallelism strategies and optimizations. This dual focus ensures that LEGOSCALE serves as a valuable tool for advancing both theoretical and systems-level research in machine learning.
>
> We have observed from 9 active GitHub issues that users are beginning to extend LEGOSCALE for fine-tuning and inference. This strongly indicates that the system’s impact extends beyond pre-training research, benefiting a broader community.
>
> **W2 and Q2**:
>
> LEGOSCALE’s key research contribution lies in analyzing and identifying a common set of design principles across parallelism and optimization techniques. These principles unify diverse methods into a cohesive system. The engineering contribution is in enforcing these principles across individual techniques, resulting in a modular, composable, and flexible distributed system that enables seamless scaling, efficient deployment, and integration of new techniques. These contributions have shaped PyTorch features like FSDP2, Async-TP, PP, and DTensor advancements.
>
> **LEGOSCALE’s Design Principles**
>
> LEGOSCALE adopts a top-down approach, enforcing modularity, non-intrusiveness, and extensibility. By leveraging DTensor and DeviceMesh, it provides a unified framework for diverse parallelisms and optimizations, ensuring scalability and compatibility with PyTorch.
>
> **Advancements to DTensor**
>
> DTensor sharding was extended to support n-D parallelism, integrating TP, CP, and PP. Compatibility with torch.compile was added for compiler optimizations, and checkpointing efficiency improved with robust state dict support. Production readiness was bolstered by resolving critical bugs.
>
> **Contribution to PyTorch FSDP2**
>
> FSDP2 replaced FlatParameter with DTensors sharded on dim-0, enabling flexible parameter handling (e.g., freezing, fp8 all-gather). It integrates with torch.compile, reduces GPU memory usage by eliminating CPU synchronization, and simplifies checkpointing with sharded state dicts.
>
> **Contribution to PyTorch Async-TP/CP**
>
> Async-TP introduced fused operators like fused_all_gather_matmul to reduce synchronization overheads, enabling fused compute and communication. This design facilitates new parallelisms like CP and minimizes bottlenecks with CUDA streams and SymmetricMemory.
>
> **Contribution to PyTorch PP**
>
> Pipeline Parallelism incorporates LEGOSCALE’s principles, enabling dynamic schedules, fine-grained stage control, and the addition of new schedules while integrating with TP and DP.
>
> **A Unified Template for Distributed Training**
>
> LEGOSCALE provides a unified design template, simplifying exploration of configurations and enabling rigorous evaluation of techniques. It unifies, modularizes, and extends distributed methods, shaping PyTorch’s library and advancing scalable, flexible systems.
>
> **W4:**
>
> We thank the reviewer for pointing this out. In the camera-ready version, we will make the text more concise to allocate space for enlarging Figures 1 and 2, ensuring better visibility and readability for the readers.

---

> ### Author Response · Authors · 2024-11-21
>
> **Q1:**
>
> Thank you for the question. We have fully open-sourced the LEGOSCALE system on GitHub (anonymized to comply with ICLR's double-anonymous requirements). The framework has garnered significant interest, with over **2,600 stars**, **204 forks**, **469 closed pull requests**, and **35 active contributors**. LEGOSCALE enables researchers and practitioners to explore, develop, and scale advanced parallelism and optimization techniques in a production-grade pre-training system.
>
> LEGOSCALE has been validated on multiple platforms, including the public cloud (AWS), on-premises clusters of a Fortune 500 company, a leading private organization, and a world-renowned university. Results are designed to be easily replicable across various cluster configurations. We are committed to actively maintaining and supporting LEGOSCALE for the foreseeable future.
>
> Since submitting the paper, we have incorporated **Context Parallelism (CP)** and added three additional **Pipeline Parallelism (PP)** schedules (Looped-BFS, Flexible-Interleaved-1F1B, ZeroBubble). The camera-ready version will include case studies showcasing how to extend LEGOSCALE and the design principles driving its modularity.
>
> Another key feature is LEGOSCALE’s model-agnostic and general design. While demonstrated with the Llama model, **LEGOSCALE is not model-specific**. Its techniques are general-purpose, allowing seamless integration of other models without changes to the system.
>
> We plan to release **Llama 3.2 multimodal recipes** soon and are working on integrating diffusion models. This would also enable research to extend beyond NLP to vision models, highlighting LEGOSCALE's flexibility and extensibility.
>
> Our goal in submitting this paper is to raise awareness within the research community about a system that empowers advanced research on pre-training large language models (LLMs) at scale, fostering innovation in this critical area.
>
> **W3**:
>
> Our experiments serve multiple objectives:
> 1. **Establish composability and modularity**: LEGOSCALE demonstrates seamless integration of various parallelisms and optimization techniques.
> 2. **Showcase performance improvements**: Significant speed-ups are observed across parallelisms and optimizations.
> 3. **Validate elastic scalability**: LEGOSCALE scales effectively with both model size and the number of GPUs.
> 4. **Ablation studies**: Detailed performance gains for individual techniques are presented:
>    - **Table 1**: Highlights improvements from compiler over eager execution, followed by gains with Float8 training.
>    - **Table 2**: Demonstrates how earlier gains scale as the number of GPUs increases.
>    - **Table 3**: Shows speed-up achieved by Async-TP over 2D training combined with `torch.compile` and Float8 training.
>    - **Table 4**: Quantifies the benefits of Interleaved 1F1B over 1F1B on top of Async-TP, `torch.compile`, and Float8 training.
>
> We acknowledge the reviewer’s concern and will address it by adding experiments on training convergence and Distributed Checkpointing (DCP) performance in the appendix to provide further scientific findings. We have already done those experiments, the graphs can be found [here](https://www.dropbox.com/scl/fo/w7pdixb5a8m8kyffycsxe/AMYZPLKvMutEfgPQSSiyDYE?rlkey=qm8i648i95b4s52eya722tej7&e=1&st=43aa066z&dl=0).

---

> > ### Comment · Reviewer_5X5L · 2024-11-24
> >
> > Thanks for your reply. The fact that the tool is already in use beyond the authors' group is a promising sign. It suggests that there may interest at the conference.

---

> > > ### Author Response · Authors · 2024-11-25
> > > **Why LEGOSCALE has seen extensive adoption? -> Novelty and broad coverage with reduced complexity**
> > >
> > > We are glad to hear that we have addressed your concerns and appreciate your recognition of LEGOSCALE’s relevance to conference attendees.
> > >
> > > ### What makes LEGOSCALE novel?
> > >
> > > **LEGOSCALE’s primary contribution lies in delivering a comprehensive distributed training system for LLMs**, rather than introducing individual distributed techniques. Its **unified design principles**—modularity, composability, and extensibility—enable seamless integration of advanced parallelism strategies and optimizations.
> > >
> > > **LEGOSCALE uniquely covers the entire design space of distributed training systems**, offering combinations of algorithms and parallelism strategies that existing systems do not support. This breadth and flexibility underscore its **novel contribution**.
> > >
> > > **Feature Comparison**
> > >
> > > - **Megatron-LM**: Requires significant model modifications and lacks advanced pipeline schedules and flexibility in combining FSDP with TP/PP.
> > > - **DeepSpeed**: Relies on Megatron-LM for TP/CP and lacks integration of advanced schedules with FSDP.
> > > - **veScale**: Lacks FSDP, CP, SAC, Float8 training, and full `torch.compile` compatibility. It supports only three pipeline schedules versus LEGOSCALE’s six and implements 3D instead of 4D parallelism.
> > >
> > >
> > > The following table highlights LEGOSCALE’s broader capabilities.
> > >
> > > | **Features**                          | **Legoscale** | **Megatron-LM**            | **DeepSpeed** | **VeScale** |
> > > |---------------------------------------|---------------|-----------------------------|---------------|-------------|
> > > | FSDP-Zero2                            | Y             | Y                           | Y             | N           |
> > > | FSDP-Zero3                            | Y             | Y                           | Y             | N           |
> > > | HSDP                                  | Y             | Y                           | N             | N           |
> > > | TP                                    | Y             | Y                           | N             | Y           |
> > > | Async TP (Micro-pipelining)           | Y             | Y                           | N             | Y           |
> > > | CP                                    | Y             | Y                           | N             | N           |
> > > | PP-Gpipe                              | Y             | Y                           | Y             | N           |
> > > | PP-Interleaved (1F1B)                 | Y             | Y                           | Y             | Y           |
> > > | PP-Looped-BFS                         | Y             | N                           | N             | N           |
> > > | PP-1F1B                               | Y             | Y                           | Y             | Y           |
> > > | PP-Flexible-Interleaved-1F1B          | Y             | N                           | N             | N           |
> > > | PP-ZeroBubble                         | Y             | N                           | N             | Y           |
> > > | (TP+SP)+PP                            | Y             | Y                           | N             | Y           |
> > > | DDP+(TP+SP)+PP                        | Y             | Y                           | N             | Y           |
> > > | FSDP +(TP + SP)                       | Y             | N                           | N             | N           |
> > > | FSDP + (TP + SP) + PP                 | Y             | N                           | N             | N           |
> > > | FSDP + (TP + SP) + PP + CP            | Y             | N                           | N             | N           |
> > > | MoE                                   | Ongoing       | Y                           | N             | N           |
> > > | Full AC                               | Y             | Y                           | Y             | Y           |
> > > | Flexible SAC                          | Y             | N                           | N             | N           |
> > > | DCP                                   | Y             | Y                           | Y             | Y           |
> > > | Float-8 Training                      | Y             | Y                           | N             | N           |
> > > | torch.compile                         | Y             | N (Custom Fusion Kernels)   | Partial       | N           |
> > >
> > > **Code Complexity:**
> > >
> > > These same design principles also contribute to LEGOSCALE’s reduced code complexity. Despite offering a rich feature set, LEGOSCALE has a compact and maintainable codebase, significantly smaller than other systems. Its modular architecture ensures extensibility and performance while minimizing complexity, making the framework easier to maintain and evolve.
> > >
> > > | **Lines of Code (LOC)**             | **Legoscale** | **Megatron-LM** | **DeepSpeed** |
> > > |-------------------------------------|---------------|------------------|---------------|
> > > | **Core Codebase**                   | 7K            | 93K             | 94K           |
> > > | **Total Codebase (Including Utils)**| 9K            | 269K            | 194K          |

---

### Official Review · Reviewer_Xmiy · 2024-11-03

**Soundness:** 3
**Presentation:** 3
**Contribution:** 3
**Rating:** 6
**Confidence:** 5

**Summary:**

The paper presents LegoScale, a framework that enables easy comparison and composition of different LLM training functionalities and optimizations. LegoScale can significantly improve the productivity of model scientists and practitioners by reducing the engineering burden of model development. LegoScale incorporates Pytorch-native implementation of popular techniques like mixed precision, 3D parallelism, ZeRO, activation checkpointing, model checkpointing, logging, etc. The evaluation studies usage of LegoScale for training Llama-3 model family and demonstrates model and hardware scaling scenarios by composing 3D parallelism, mixed precision, and compiler optimizations.

**Strengths:**

- LegoScale tackles an important problem of making it easier for model practitioners and researchers to compare and compose SOTA training techniques. Advanced LLM training techniques are typically scattered across different frameworks making it difficult if not impractical for practitioners and researchers to leverage these techniques. LegoScale could be a very valuable tool for the community.
- By incorporating pytorch-native implementations of training optimizations, LegoScale lowers the bar for integration into existing pytorch workflows, likely fostering adoption.
- The evaluation results highlight intuitive compositions of relevant optimizations for model and hardware scaling, which helps the user appreciate the developer benefits of LegoScale.

**Weaknesses:**

- Considering the growing importance of context scaling, to millions of tokens, LegoScale seems to be missing an important scenario. Although, this is discussed as ongoing work, it is unclear how context scaling fits into LegoScale. This is because the current tensor abstraction focuses on model partitioning, whereas SOTA context scaling optimizations such as Ulyssess involve input partitioning.
- While LegoScale allows composition of various optimizations, it is unclear how users will be guided to make optimal choices in terms of performance metrics such as throughput, latency, or hardware utilization.
- While the evaluation does a good presenting the scaling performance results, it is missing results of the production-ready-training features (2.3). For example, it would be useful to show convergence plots demonstrating that effectiveness of parallelism changes using DCP.
- The performance evaluation lacks a common baseline metric across Tables 1-4, which makes it difficult to understand LegoScale efficiency across model and hardware scaling dimensions. While the authors rightly identified how different precisions hampers MFU usage, I think a good compromise is to report bfloat16 and float8 MFUs separately.

**Questions:**

1. Can you clarify whether the throughput results in Tables 1-4 are per GPU or aggerate.
2. How are the optimization configurations (e.g., PP vs FSDP, AC vs SAC) used in Tables 1-4 obtained? Was there some exploration of the configuration space to discover optimal settings?
3. Can you clarify that LegoScale applies to other model families besides Llama? In particular, line 279 refers to Llama model when discussing the torch.compile integration (2.2.2).
4. What ZeRO stages are the FSDP configurations in Tables 1-4?
5. HSDP is not mentioned in Figure 1 or 2, although mentioned as an important FSDP option. Is HSDP not available in LegoScale?

---

> ### Author Response · Authors · 2024-11-21
>
> We sincerely thank the reviewer for underscoring the importance of our work, recognizing our contributions, appreciating the design of our system, and providing detailed and critical feedback on our work. Please consider our response labeled as Wi, which aims to address weaknesses i, while Qi aims to address Question i, pointed out by the reviewer.
>
> **W1**:
>
> We thank the reviewer for highlighting the importance of context scaling and its integration into LEGOSCALE. We are pleased to report that, since submitting the paper, we have successfully incorporated **Context Parallelism (CP)** into LEGOSCALE. We will include details of this in the camera-ready version of the paper. Below, we provide an outline of how LEGOSCALE addresses context scaling through CP.
>
> ### Tensor Parallelism (TP) with DTensor
> LEGOSCALE leverages **DTensor** to shard not only model parameters but also inputs and activations along multiple dimensions. For example, in an MLP layer:
> - The first linear layer’s parameters are **column-sharded**, while the second layer’s parameters are **row-sharded**.
> - Inputs are sharded across the batch and feature dimensions, enabling flexible, efficient handling of sharded computations.
>
> This design ensures that TP is not limited to parameter sharding and can effectively scale across different workloads.
>
> ### Context Parallelism (CP) in LEGOSCALE
> To address **context scaling**, LEGOSCALE extends DTensor’s capabilities to support CP. Following LEGOSCALE’s modular design principles, CP was integrated using two user-friendly APIs:
> 1. **Module Wrapper**: Similar to TP, for distributing the Attention module.
> 2. **Context Manager**: Dynamically replaces `scaled_dot_product_attention` calls with context-parallel operators, ensuring no changes to the model code.
>
> Under the hood, CP:
> - Shards the **DTensor** along the sequence dimension across the CP device mesh.
> - Extends the DTensor dispatcher to handle **context-parallel-specific operations**, such as **ring-attention** and **causal attention load balancing**, ensuring efficient operation.
>
> By extending DTensor’s design, CP in LEGOSCALE remains fully compatible with other parallelisms (FSDP, TP, PP), optimizations (e.g., activation checkpointing, `torch.compile`), and **Distributed Checkpointing (DCP)**. This demonstrates LEGOSCALE’s extensibility and future-proof design, accommodating new optimizations seamlessly while maintaining performance and compatibility.
>
> ### Context Scaling Results (Llama3-8B, 64 H100 GPUs, TP=8, DPxCP=8)
>
> | **CP Degree** (GPUs on CP dimension) | **Max Context Length** (Without OOM) | **Local WPS** (linear scaling) | **MFU** (slight degradation) |
> |---------------------------------------|---------------------------------------|--------------------------------|-------------------------------|
> | 1                                     | 32,768                                | 4497                           | 43.09                         |
> | 2                                     | 81,920                                | 1970                           | 34.63                         |
> | 4                                     | 147,456                               | 1205                           | 33.73                         |
> | 8                                     | 294,912                               | 638                            | 34.09                         |
>
> The integration of Context Parallelism highlights LEGOSCALE’s ability to adapt to emerging requirements in LLM training. Its unified DTensor abstraction and modular framework ensure that context scaling and other advancements can be seamlessly incorporated, making it a robust solution for future research and deployment.
>
> **W2:**
>
> We appreciate the reviewer’s insightful question. Identifying the best configuration parameters for a 4D parallel system, along with compute and memory optimizations, requires a **modular, composable, and extensible framework** as a foundation. This is the primary contribution of LEGOSCALE—providing a unified system that enables users to systematically explore and evaluate complex configurations.
>
> Optimizing performance involves efficiently navigating a design space influenced by factors such as checkpointing strategies (AC vs. SAC), parallelism degrees (DP/CP/TP/PP), memory layouts (HSDP vs. FSDP), and pipeline schedules (e.g., 1F1B, Zero-Bubble). This requires estimating GPU compute times, communication latencies, and memory footprints, followed by algorithms that recommend a configuration, which presents a new research direction.
>
> To assist users, we provide practical guidelines on choosing the parallelism strategies in **Section 3.3**.  Additionally, we are prototyping tools like **AutoSAC** and **AutoFSDP** to automate configuration tuning.
>
> While fully solving the optimization problem remains an open challenge, LEGOSCALE offers the foundation researchers need to experiment and develop solutions for distributed training.

---

> > ### Author Response · Authors · 2024-11-21
> >
> > **W3:**:
> >
> >
> > We thank the reviewer for highlighting the importance of presenting convergence results to demonstrate the effectiveness of LEGOSCALE’s production-ready features. LEGOSCALE’s design principles have influenced the development of advanced distributed training features such as FSDP2, Async-TP, and PP in PyTorch’s distributed library. In making these contributions, we ensured that loss convergence is maintained across individual techniques and combinations of parallelisms and optimizations.
> >
> > We will include convergence results in the appendix. For now, loss curves for **Llama-3 (8B/70B, 1D/2D with Float8 and `torch.compile`)** training can be accessed [here](https://www.dropbox.com/scl/fo/w7pdixb5a8m8kyffycsxe/AMYZPLKvMutEfgPQSSiyDYE?rlkey=qm8i648i95b4s52eya722tej7&e=1&st=43aa066z&dl=0).
> >
> > **W4:**
> >
> > Our experiments serve multiple objectives:
> > 1. **Establish composability and modularity**: LEGOSCALE demonstrates seamless integration of various parallelisms and optimization techniques.
> > 2. **Showcase performance improvements**: Significant speed-ups are observed across parallelisms and optimizations.
> > 3. **Validate elastic scalability**: LEGOSCALE scales effectively with both model size and the number of GPUs.
> > 4. **Ablation studies**: Detailed performance gains for individual techniques are presented:
> >    - **Table 1**: Highlights improvements from compiler over eager execution, followed by gains with Float8 training.
> >    - **Table 2**: Demonstrates how earlier gains scale as the number of GPUs increases.
> >    - **Table 3**: Shows speed-up achieved by Async-TP over 2D training combined with `torch.compile` and Float8 training.
> >    - **Table 4**: Quantifies the benefits of Interleaved 1F1B over 1F1B on top of Async-TP, `torch.compile`, and Float8 training.
> >
> > As discussed in **Section 3.3 (Scaling with LegoScale 3D parallelism)**, the combinations of parallelisms (e.g., 1D, 2D, 3D) truly showcase their power in LLM training when the scale reaches **1k–4k GPUs**. Unfortunately, we did not have access to such resources during benchmarking. Instead, we addressed this question by analyzing the characteristics and scalability of each parallelism strategy in the paper.
> >
> > With mixed precision training that combines Float8 (for linear layers) and BFloat16 (for other operations), accurately reporting MFU becomes challenging due to the differing peak FLOPs for tensor cores depending on precision.
> >
> > For experiments without Float8 training, we reported the following in the paper:
> > > *“We note that the 1D Llama 3.1 8B model training on 8 or 128 H100 GPUs without Float8 achieves 33% to 42% MFU.”*
> >
> > The improvement from 33% to 42% primarily comes from `torch.compile`.
> >
> > **Q1:**
> >
> > Thank you for your question. The throughput results in Tables 1-4 are **per GPU**, as noted in **Line 388, Section 3.2**:
> > > *"Throughput numbers (tokens per second, per GPU) are calculated and logged every 10 iterations and read at the 90th iteration."*
> >
> > We appreciate the suggestion and will clarify this further by including the information in the captions for each table.
> >
> > **Q2:**
> >
> >  Yes, we performed an experimental study by sweeping across a subset of configuration options to determine the settings used in Tables 1-4. To further assist users, we have incorporated a **memory estimation tool** into LEGOSCALE. This tool helps narrow down configuration options, enabling more efficient benchmarking and guiding users toward discovering optimal settings.
> >
> > **Q3:**
> >
> > This is a great question that highlights LEGOSCALE’s model-agnostic and general design. While demonstrated with the Llama model, **LEGOSCALE is not model-specific**. Its techniques are general-purpose, allowing seamless integration of other models without changes to the system.
> >
> > We plan to release **Llama 3.2 multimodal recipes** soon and are working on integrating diffusion models, showcasing LEGOSCALE’s flexibility and extensibility.
> >
> > **Q4:**
> >
> > Thank you for the suggestion. Tables 1-3 use **ZeRO-3** for 1D and 2D experiments, while **ZeRO-2** is used for experiments involving PP. This distinction is due to the inefficiency of ZeRO-3 in PP, where it incurs additional all-gather calls for each micro-batch. In contrast, ZeRO-2 gathers parameters only once for the first micro-batch and reshards after the last micro-batch’s backward pass.
> >
> > We will include this detail in the experimental analysis section to provide greater clarity.
> >
> > **Q5:**
> >
> > Thank you for pointing this out. Although HSDP is not explicitly mentioned in Figures 1 and 2, it is fully supported in LEGOSCALE. A detailed description is provided in **Appendix B2** of the paper. Here is a summary:
> >
> > HSDP configurations are defined using two parameters: **dp_replicate_degree** and **dp_shard_degree**. Setting `dp_shard_degree = dp_degree` corresponds to FSDP, while `dp_replicate_degree > 1` enables HSDP with replicas equal to `dp_replicate_degree`.
> >
> > We will ensure these details are included in the final version of the paper for clarity.

---

### Official Review · Reviewer_izCA · 2024-11-04

**Soundness:** 3
**Presentation:** 3
**Contribution:** 3
**Rating:** 6
**Confidence:** 3

**Summary:**

This paper introduces `LegoScale`, an open-source, PyTorch-native distributed training system designed to unify and enhance current engineering efforts in LLM pre-training. `LegoScale` improves training efficiency through a modular, composable design that supports various parallelism techniques, including Fully Sharded Data Parallel (FSDP), Tensor Parallel (TP), and Pipeline Parallel (PP), as well as memory optimization methods such as FP8 precision and activation checkpointing. It also integrates with `torch.compile` for further optimizations. Experiments on the Llama 3.1 family of LLMs demonstrate that `LegoScale` accelerates training by 65.08% on the Llama 3.1 8B model at a 128-GPU scale (1D), 12.59% on the Llama 3.1 70B model at a 256-GPU scale (2D), and 30% on the Llama 3.1 405B model at a 512-GPU scale (3D), all on NVIDIA H100 GPUs, compared to optimized baselines.

**Strengths:**

This paper is well-organized and clearly presents the design of  `LegoScale`  across parallelism strategies, memory optimization techniques, and integration with  `torch.compile`. While prior research has addressed various aspects of LLM training, this paper delivers a comprehensive, feature-complete open-source solution that focuses on key challenges, including:

1.  Composability of parallelism techniques,
2.  Extensibility for incorporating new optimizations,
3.  Efficient hardware utilization, and
4.  Production-ready features.

The figures, particularly Figures 1 and 2, effectively illustrate how the system supports parallelism and customization features.

Notably, the authors provide clear abstractions for N-D parallelism in training and analyze strategies for maximizing efficiency when combining parallelism methods. The experiments demonstrate promising performance improvements compared to strong baselines. Beyond these contributions, the engineering effort invested in this project is substantial and valuable for future work in this area.

**Weaknesses:**

The proposed training system builds extensively on prior work, especially in parallelism design and memory optimization. While the tensor abstraction is novel and more expressive, the primary contribution is from an engineering perspective. Additionally, possibly due to page limitations, much of the detail is relegated to the appendix. There are a few areas where the paper could strengthen its credibility.

First, the paper could benefit from more discussion on how the modular design facilitates the integration of future optimizations and enables further parallelism without compromising performance.

For the PyTorch integration, although  `torch.compile`  is powerful, the paper lacks specifics on how it sustains high GPU efficiency in a highly parallel training environment—for example, insights on GPU usage.

An ablation study on the effect of individual features, such as transitioning from FSDP1 to FSDP2, and on performance changes as GPU scale increases within 1D, 2D, and 3D parallelism settings, would also help readers better understand the system's effectiveness.

**Minor issue:**  In line 315, "torch" is misspelled as "torchao."

**Questions:**

-   Could you elaborate on the improvements from FSDP1 to FSDP2, and how these enhancements impact different parallelism settings?
-   How does each parallelism strategy scale with the number of GPUs?
-   What does GPU utilization look like during training? Is GPU efficiency maximized?
-   Is this method extensible to other platforms, such as TPU clusters?
-   Could you provide more details on the system's fault tolerance and crash recovery design?

---

> ### Author Response · Authors · 2024-11-21
>
> We sincerely thank the reviewer for acknowledging the strengths of our work and highlighting them. We appreciate the detailed and critical feedback on our work. Please consider our response labeled as Wi, which aims to address weaknesses i, while Qi aims to address Question i, pointed out by the reviewer.
>
>
> **W1**:
>
> As the reviewer points out, many techniques used in LEGOSCALE are well-documented. However, LEGOSCALE’s key research contribution lies in analyzing and identifying a common set of design principles across parallelism and optimization techniques. These principles unify diverse methods into a cohesive system. The engineering contribution is in enforcing these principles across individual techniques, resulting in a modular, composable, and flexible distributed system that enables seamless scaling, efficient deployment, and integration of new techniques. These contributions have shaped PyTorch features like FSDP2, Async-TP, PP, and DTensor advancements.
>
> **LEGOSCALE’s Design Principles**
>
> LEGOSCALE adopts a top-down approach, enforcing modularity, non-intrusiveness, and extensibility. By leveraging DTensor and DeviceMesh, it provides a unified framework for diverse parallelisms and optimizations, ensuring scalability and compatibility with PyTorch.
>
> **Advancements to DTensor**
>
> DTensor sharding was extended to support n-D parallelism, integrating TP, CP, and PP. Compatibility with torch.compile was added for compiler optimizations, and checkpointing efficiency improved with robust state dict support. Production readiness was bolstered by resolving critical bugs.
>
> **Contribution to PyTorch FSDP2**
>
> FSDP2 replaced FlatParameter with DTensors sharded on dim-0, enabling flexible parameter handling (e.g., freezing, fp8 all-gather). It integrates with torch.compile, reduces GPU memory usage by eliminating CPU synchronization, and simplifies checkpointing with sharded state dicts.
>
> **Contribution to PyTorch Async-TP/CP**
>
> Async-TP introduced fused operators like fused_all_gather_matmul to reduce synchronization overheads, enabling fused compute and communication. This design facilitates new parallelisms like CP and minimizes bottlenecks with CUDA streams and SymmetricMemory.
>
> **Contribution to PyTorch PP**
>
> Pipeline Parallelism incorporates LEGOSCALE’s principles, enabling dynamic schedules, fine-grained stage control, and the addition of new schedules while integrating with TP and DP.
>
> **A Unified Template for Distributed Training**
>
> LEGOSCALE provides a unified design template, simplifying exploration of configurations and enabling rigorous evaluation of techniques. It unifies, modularizes, and extends distributed methods, shaping PyTorch’s library and advancing scalable, flexible systems.
>
>
> **W2:**
>
> We thank the reviewer for their insightful comment on how LEGOSCALE’s modular design supports future optimizations and parallelism strategies without compromising performance. Since submitting the paper, we have integrated **Context Parallelism (CP)** and added three new **Pipeline Parallelism (PP)** schedules: Looped-BFS, Flexible-Interleaved-1F1B, and ZeroBubble. The camera-ready version will include case studies demonstrating how to extend LEGOSCALE and its design principles. Below, we summarize how CP was enabled.
>
> ### Context Parallelism (CP) Integration
> LEGOSCALE integrates CP through:
> 1. **Module Wrapper**: Distributes the Attention module, similar to TP.
> 2. **Context Manager**: Dynamically replaces `scaled_dot_product_attention` with context-parallel operators, requiring no model code changes.
>
> Under the hood, CP shards **DTensor** along the sequence dimension across the CP device mesh. It extends DTensor’s dispatcher to support context-specific operations like **ring-attention** and **causal attention load balancing**, ensuring efficient operation. LEGOSCALE ensures full compatibility of CP with FSDP, TP, PP, optimizations (e.g., `torch.compile`), and DCP, showcasing its extensibility and seamless integration of new techniques.
>
> ### Context Scaling Results (Llama3-8B, 64 H100 GPUs, TP=8, DPxCP=8)
>
> | **CP Degree** (GPUs on CP dimension) | **Max Context Length** (Without OOM) | **Local WPS** (linear scaling) | **MFU** (slight degradation) |
> |---------------------------------------|---------------------------------------|--------------------------------|-------------------------------|
> | 1                                     | 32,768                                | 4497                           | 43.09                         |
> | 2                                     | 81,920                                | 1970                           | 34.63                         |
> | 4                                     | 147,456                               | 1205                           | 33.73                         |
> | 8                                     | 294,912                               | 638                            | 34.09                         |

---

> > ### Author Response · Authors · 2024-11-21
> >
> > **Q1:**
> >
> > We provide details in **Appendix B1** of the paper, and a summary is as follows:
> >
> > The authors actively contributed to the redesign of **FSDP1** into **FSDP2**, addressing key limitations. Replacing `FlatParameter` with **DTensors sharded on dim-0** enabled flexible parameter handling (e.g., freezing and fp8 all-gather). FSDP2 integrates seamlessly with parallelisms like **TP, CP, and PP**, and tools like `torch.compile`. It improves GPU memory efficiency by eliminating CPU synchronization (e.g., `recordStream`) and simplifies checkpointing with **communication-free, sharded state dicts** aligned with training representations.
> >
> > These enhancements ensure better memory efficiency, composability, and flexibility across diverse parallelism settings.
> >
> > **Q2 and W4:**
> >
> > Our experiments serve multiple objectives:
> > 1. **Establish composability and modularity**: LEGOSCALE demonstrates seamless integration of various parallelisms and optimization techniques.
> > 2. **Showcase performance improvements**: Significant speed-ups are observed across parallelisms and optimizations.
> > 3. **Validate elastic scalability**: LEGOSCALE scales effectively with both model size and the number of GPUs.
> > 4. **Ablation studies**: Detailed performance gains for individual techniques are presented:
> >    - **Table 1**: Highlights improvements from compiler support over eager execution, followed by further gains with Float8 training.
> >    - **Table 2**: Demonstrates how earlier gains scale as the number of GPUs increases.
> >    - **Table 3**: Shows speed-up achieved by Async-TP (a HW/SW co-designed technique) over 2D training combined with `torch.compile` and Float8 training.
> >    - **Table 4**: Quantifies the benefits of Interleaved 1F1B scheduling over 1F1B on top of Async-TP, `torch.compile`, and Float8 training.
> >
> > As discussed in **Section 3.3 (Scaling with LegoScale 3D parallelism)**, the combinations of parallelisms (e.g., 1D, 2D, 3D) truly showcase their power in LLM training when the scale reaches **1k–4k GPUs**. Unfortunately, we did not have access to such resources during benchmarking. Instead, we addressed this question by analyzing the characteristics and scalability of each parallelism strategy in the paper.
> >
> > **Q3 and W3:**
> >
> > Thank you for the insightful question regarding GPU utilization and efficiency. GPU utilization is typically measured via Maximum FLOP Utilization (MFU). However, with mixed precision training that combines Float8 (for linear layers) and BFloat16 (for other operations), accurately reporting MFU becomes challenging due to the differing peak FLOPs for tensor cores depending on precision.
> >
> > For experiments without Float8 training, we reported the following in the paper:
> > > *“We note that the 1D Llama 3.1 8B model training on 8 or 128 H100 GPUs without Float8 achieves 33% to 42% MFU.”*
> >
> > The improvement from 33% to 42% primarily comes from the optimizations introduced by `torch.compile`.
> >
> > We would greatly appreciate it if the reviewer could suggest a methodology or best practice for measuring MFU in mixed precision settings. Such guidance would be invaluable in further refining our analysis and providing clearer insights into GPU efficiency for highly parallel training environments.
> >
> >
> > **Q4:**
> >
> > We appreciate the reviewer’s insightful question regarding the extensibility of LEGOSCALE to heterogeneous platforms such as TPU clusters. Specifically, PyTorch provides the **PyTorch/XLA** package, which bridges the PyTorch framework with Cloud TPUs using the XLA deep learning compiler. With the release of PyTorch/XLA 2.0, experimental support for `torch.compile` was introduced, enabling TorchDynamo to compile TorchFX graphs for both inference and training workloads on TPUs.
> >
> > Since LEGOSCALE is fully PyTorch-native and supports `torch.compile`, it can directly utilize the PyTorch/XLA backend for both eager execution and compiler-driven workflows. For distributed training on TPUs, PyTorch/XLA leverages **GSPMD** (General and Scalable Parallel Model Development), an automatic parallelization system in the XLA compiler. GSPMD transforms single-device programs into partitioned ones with appropriate collectives, guided by user-provided sharding hints.
> >
> > Importantly, there is a **1:1 mapping** between PyTorch’s DTensor, DeviceMesh, and Placement abstractions and GSPMD’s DistributedTensor, Mesh, and PartitionSpec. This alignment ensures that forking LEGOSCALE to leverage GSPMD for TPU workflows is straightforward and preserves its modularity, composability, and extensibility. Such a port would highlight LEGOSCALE’s robust design, enabling comparisons across heterogeneous cloud architectures while bringing the benefits of XLA compilation and GSPMD to PyTorch users.
> >
> > For further details: [PyTorch/XLA Documentation](https://pytorch.org/xla/master/index.html)

---

> > > ### Author Response · Authors · 2024-11-21
> > >
> > > **Q5:**
> > >
> > >
> > > Thank you for the crucial question. The mechanisms for fault tolerance and crash recovery are detailed in **Section 2.3** of the paper, and we provide a summary below for clarity:
> > >
> > > 1. **Checkpointing with DCP**:
> > >    LEGOSCALE enhances PyTorch Distributed Checkpointing (DCP) by integrating it with **DTensor primitives**, enabling seamless checkpointing across TP, CP, PP, and FSDP. DTensor standardizes shard representations, ensuring compatibility across parallelisms. To minimize overhead, LEGOSCALE supports:
> > >    - **Asynchronous checkpointing**: Overlaps storage operations with training iterations, reducing checkpoint time by **19x** compared to synchronous methods.
> > >    - **Zero-overhead checkpointing**: Overlaps GPU-to-CPU transfers with computation, achieving an additional **5x reduction**, bringing total checkpoint times to under one second.
> > >
> > > 2. **Debugging with Flight Recorder**:
> > >    PyTorch’s Flight Recorder simplifies debugging large-scale training hangs or crashes, such as NCCL collective timeouts. It logs GPU start/end times, CPU enqueue times, tensor sizes, process groups, source/destination ranks, and stack traces for collectives and point-to-point operations. This helps identify issues like missing or misordered sends/receives in PP schedules or rank mismatches in FSDP/TP. By enabling Flight Recorder, LEGOSCALE provides users a powerful tool to diagnose and resolve parallelism workflow failures.
> > >
> > > Together, these features provide a robust fault-tolerance framework, minimizing downtime and simplifying recovery in distributed training scenarios. We provide performance graphs for DCP here: [DCP-Performance](https://www.dropbox.com/scl/fo/w7pdixb5a8m8kyffycsxe/AMYZPLKvMutEfgPQSSiyDYE?rlkey=qm8i648i95b4s52eya722tej7&st=43aa066z&dl=0)
> > >
> > > **Minor Issue**
> > >
> > > Thank you for pointing this out. To clarify, “torchao” refers to the **PyTorch Architecture Optimization (AO)** library, which is designed to support custom data types and optimizations. It enables users to quantize and sparsify weights, gradients, optimizers, and activations for both inference and training.
> > >
> > > We will add clarification regarding this in the camera-ready version of the paper.
> > >
> > > For more details, please refer to the [PyTorch AO GitHub repository](https://github.com/pytorch/ao).

---

> > > > ### Comment · Reviewer_izCA · 2024-11-24
> > > > **Response to Rebuttal**
> > > >
> > > > Thank you for your detailed rebuttal—it has addressed most of my concerns effectively. I appreciate the significant engineering effort demonstrated in the work. However, I regret that I am unable to raise my score further due to the limited novelty.

---

> > > > > ### Author Response · Authors · 2024-11-25
> > > > > **Novelty 1/2**
> > > > >
> > > > > We are glad to hear that we have effectively addressed your concerns, and we appreciate your recognition of the significant engineering effort demonstrated in our work. Regarding the question of novelty, we would like to reiterate that **LEGOSCALE’s primary contribution lies in delivering a complete distributed training system for LLMs**, rather than introducing individual distributed techniques.
> > > > >
> > > > > LEGOSCALE’s **unified design principles**—modularity, composability, and extensibility—enable seamless integration of FSDP, TP, PP, CP, and optimizations like SAC and Float8 training. This system abstracts away the complexities of distributed systems, allowing **machine learning practitioners** to focus on architectural exploration, training dynamics, and mathematical problems. Simultaneously, LEGOSCALE serves as a **testbed for ML systems researchers**, enabling them to invent, extend, and seamlessly integrate new techniques across parallelism strategies and optimizations. This dual focus bridges theoretical and systems-level research, making LEGOSCALE a valuable tool for advancing machine learning.
> > > > >
> > > > > We have fully open-sourced the LEGOSCALE system on GitHub (anonymized to comply with ICLR's double-anonymous requirements), where it has garnered significant interest, including:
> > > > > - **2,600+ stars**
> > > > > - **204 forks**
> > > > > - **469 closed pull requests**
> > > > > - **35 active contributors**
> > > > >
> > > > > This level of community engagement highlights its value as a production-grade pre-training system for exploring, developing, and scaling distributed training techniques.
> > > > >
> > > > > We also emphasize that **LEGOSCALE is the only system to comprehensively cover the entire design space of distributed training systems**. Compared to related systems like Megatron-LM, DeepSpeed, and veScale, LEGOSCALE uniquely offers combinations of algorithms and parallelism strategies that none of these systems support. This breadth and flexibility underscore its **novel contribution** to the field.
> > > > >
> > > > > **Feature Comparison**
> > > > >
> > > > > LEGOSCALE addresses key limitations:
> > > > >
> > > > > - Megatron-LM: Requires significant model modifications and lacks advanced pipeline schedules and flexibility in combining FSDP with TP/PP.
> > > > >
> > > > > - DeepSpeed: Relies on Megatron-LM for TP/CP and lacks integration of advanced schedules with FSDP.
> > > > >
> > > > > - veScale: Does not support FSDP, CP, SAC, Float8 training, or full torch.compile compatibility. It offers three pipeline schedules compared to LEGOSCALE’s six and treats SP as a separate dimension, implementing 3D instead of 4D parallelism.
> > > > >
> > > > >
> > > > > We hope this perspective helps highlight the uniqueness and impact of LEGOSCALE.

---

> > > > > ### Author Response · Authors · 2024-11-25
> > > > > **Novelty 2/2**
> > > > >
> > > > > The following table highlights LEGOSCALE’s broader capabilities.
> > > > >
> > > > > | **Features**                          | **Legoscale** | **Megatron-LM**            | **DeepSpeed** | **VeScale** |
> > > > > |---------------------------------------|---------------|-----------------------------|---------------|-------------|
> > > > > | FSDP-Zero2                            | Y             | Y                           | Y             | N           |
> > > > > | FSDP-Zero3                            | Y             | Y                           | Y             | N           |
> > > > > | HSDP                                  | Y             | Y                           | N             | N           |
> > > > > | TP                                    | Y             | Y                           | N             | Y           |
> > > > > | Async TP (Micro-pipelining)           | Y             | Y                           | N             | Y           |
> > > > > | CP                                    | Y             | Y                           | N             | N           |
> > > > > | PP-Gpipe                              | Y             | Y                           | Y             | N           |
> > > > > | PP-Interleaved (1F1B)                 | Y             | Y                           | Y             | Y           |
> > > > > | PP-Looped-BFS                         | Y             | N                           | N             | N           |
> > > > > | PP-1F1B                               | Y             | Y                           | Y             | Y           |
> > > > > | PP-Flexible-Interleaved-1F1B          | Y             | N                           | N             | N           |
> > > > > | PP-ZeroBubble                         | Y             | N                           | N             | Y           |
> > > > > | (TP+SP)+PP                            | Y             | Y                           | N             | Y           |
> > > > > | DDP+(TP+SP)+PP                        | Y             | Y                           | N             | Y           |
> > > > > | FSDP +(TP + SP)                       | Y             | N                           | N             | N           |
> > > > > | FSDP + (TP + SP) + PP                 | Y             | N                           | N             | N           |
> > > > > | FSDP + (TP + SP) + PP + CP            | Y             | N                           | N             | N           |
> > > > > | MoE                                   | Ongoing       | Y                           | N             | N           |
> > > > > | Full AC                               | Y             | Y                           | Y             | Y           |
> > > > > | Flexible SAC                          | Y             | N                           | N             | N           |
> > > > > | DCP                                   | Y             | Y                           | Y             | Y           |
> > > > > | Float-8 Training                      | Y             | Y                           | N             | N           |
> > > > > | torch.compile                         | Y             | N (Custom Fusion Kernels)   | Partial       | N           |
> > > > >
> > > > > **Code Complexity:**
> > > > >
> > > > > These same design principles also contribute to LEGOSCALE’s reduced code complexity. Despite offering a rich feature set, LEGOSCALE has a compact and maintainable codebase, significantly smaller than other systems. Its modular architecture ensures extensibility and performance while minimizing complexity, making the framework easier to maintain and evolve.
> > > > >
> > > > > | **Lines of Code (LOC)**             | **Legoscale** | **Megatron-LM** | **DeepSpeed** |
> > > > > |-------------------------------------|---------------|------------------|---------------|
> > > > > | **Core Codebase**                   | 7K            | 93K             | 94K           |
> > > > > | **Total Codebase (Including Utils)**| 9K            | 269K            | 194K          |
> > > > >
> > > > > LEGOSCALE achieves comparable or superior functionality with a fraction of the codebase, making it easier to maintain, extend, and integrate.

---

### Official Review · Reviewer_3H4A · 2024-11-05

**Soundness:** 2
**Presentation:** 2
**Contribution:** 2
**Rating:** 5
**Confidence:** 4

**Summary:**

The proposed LEGOSCALE framework aims to enhance the training efficiency of the Llama 3.1 model in distributed training by integrating several established techniques, including n-D parallelism support, efficient checkpoint, compiler enhancement, debugging tools, and Float8 support.
Authors evaluate the scaling performance of LEGOSCALE using the Llama 3.1 family of models (8B, 70B, and 405B) on scales from 8 to 512 H100 GPUs.
The paper claims 65.08%, 12.59%, and 30% improvement on training speed on the 8B model with data parallel, 70B model with data- and tensor-parallel, and 405B model with 3D parallel, repsectively.

**Strengths:**

1. Using the latest Llama 3.1 family of models is promising
2. The scale with up to 512 H100 GPUs, though about two orders of magnitude smaller than the original Llama 3.1 paper, is sufficient.

**Weaknesses:**

1. The primary concern is that none of the proposed techniques is new or invented by the authors, significantly limiting LEGOSCALE's originality. Most of the performance improvement in Table 1-4 is attributed to the incorporation of existing solutions, not the new design of LEGOSCALE.
2. LEGOSCALE employs a composable multi-dimensional parallelism approach by combining Fully Sharded Data Parallel (FSDP), Tensor Parallel (TP), and Pipeline Parallel (PP) methods, which have been extensively validated in systems like DeepSpeed (https://github.com/microsoft/DeepSpeed), Megatron-LM, and GPipe for efficient model scaling.
3. Additionally, LEGOSCALE leverages hardware-software co-designed solutions such as Float8 mixed precision and optimized activation checkpointing to improve hardware utilization and memory efficiency. Most of the recent LLM training was done with BF16 and FP32. FP8 was rarely used due to practical concerns in optimization stability. To justify the innovation of FP8, the authors need to provide end-to-end training results and downstream task performance.
4. Furthermore, LEGOSCALE incorporates PyTorch’s Distributed Checkpointing (DCP) to enable efficient failure recovery and state persistence, building upon checkpointing strategies already prevalent in systems like ByteCheckpoint and Gemini.
5. LEGOSCALE optimizes memory and computation efficiency through `torch.compile` regional compilation, a technique similarly found in frameworks like JAX and ONNX, which use compiler optimizations to enhance training throughput and resource utilization.

**Questions:**

Please address the points in weakness.

---

> ### Author Response · Authors · 2024-11-21
>
> We thank the reviewer for providing detailed and critical feedback on our work. Please consider our response labeled as Wi, which aims to address weaknesses i, pointed out by the reviewer.
>
> **W1**:
>
> As the reviewer points out, many techniques used in LEGOSCALE are well-documented. However, LEGOSCALE’s key research contribution lies in analyzing and identifying a common set of design principles across parallelism and optimization techniques. These principles unify diverse methods into a cohesive system. The engineering contribution is in enforcing these principles across individual techniques, resulting in a modular, composable, and flexible distributed system that enables seamless scaling, efficient deployment, and integration of new techniques. These contributions have shaped PyTorch features like FSDP2, Async-TP, PP, and DTensor advancements.
>
> **LEGOSCALE’s Design Principles**
>
> LEGOSCALE adopts a top-down approach, enforcing modularity, non-intrusiveness, and extensibility. By leveraging DTensor and DeviceMesh, it provides a unified framework for diverse parallelisms and optimizations, ensuring scalability and compatibility with PyTorch.
>
> **Advancements to DTensor**
>
> DTensor sharding was extended to support n-D parallelism, integrating TP, CP, and PP. Compatibility with torch.compile was added for compiler optimizations, and checkpointing efficiency improved with robust state dict support. Production readiness was bolstered by resolving critical bugs.
>
> **Contribution to PyTorch FSDP2**
>
> FSDP2 replaced FlatParameter with DTensors sharded on dim-0, enabling flexible parameter handling (e.g., freezing, fp8 all-gather). It integrates with torch.compile, reduces GPU memory usage by eliminating CPU synchronization, and simplifies checkpointing with sharded state dicts.
>
> **Contribution to PyTorch Async-TP/CP**
>
> Async-TP introduced fused operators like fused_all_gather_matmul to reduce synchronization overheads, enabling fused compute and communication. This design facilitates new parallelisms like CP and minimizes bottlenecks with CUDA streams and SymmetricMemory.
>
> **Contribution to PyTorch PP**
>
> Pipeline Parallelism incorporates LEGOSCALE’s principles, enabling dynamic schedules, fine-grained stage control, and the addition of new schedules while integrating with TP and DP.
>
> A Unified Template for Distributed Training
>
> LEGOSCALE provides a unified design template, simplifying exploration of configurations and enabling rigorous evaluation of techniques. It unifies, modularizes, and extends distributed methods, shaping PyTorch’s library and advancing scalable, flexible systems.

---

> > ### Author Response · Authors · 2024-11-21
> >
> > W2:
> >
> > We thank the reviewer for their comments and will include a detailed discussion on related work in the paper to contextualize LEGOSCALE’s contributions relative to DeepSpeed, Megatron-LM, and veScale.
> >
> > LEGOSCALE’s unified design principles—modularity, composability, and extensibility—enable seamless integration of FSDP, TP, PP, CP, and optimizations like SAC and Float8 training. This framework supports advanced pipeline schedules, multi-dimensional parallelism, and simplifies integration of new techniques for research and production.
> >
> > Feature Comparison
> >
> > LEGOSCALE addresses key limitations:
> >
> > Megatron-LM: Requires significant model modifications and lacks advanced pipeline schedules and flexibility in combining FSDP with TP/PP.
> >
> > DeepSpeed: Relies on Megatron-LM for TP/CP and lacks integration of advanced schedules with FSDP.
> >
> > veScale: Does not support FSDP, CP, SAC, Float8 training, or full torch.compile compatibility. It offers three pipeline schedules compared to LEGOSCALE’s six and treats SP as a separate dimension, implementing 3D instead of 4D parallelism.
> >
> > The following table highlights LEGOSCALE’s broader capabilities.
> >
> > | **Features**                          | **Legoscale** | **Megatron-LM**            | **DeepSpeed** | **VeScale** |
> > |---------------------------------------|---------------|-----------------------------|---------------|-------------|
> > | FSDP-Zero2                            | Y             | Y                           | Y             | N           |
> > | FSDP-Zero3                            | Y             | Y                           | Y             | N           |
> > | HSDP                                  | Y             | Y                           | N             | N           |
> > | TP                                    | Y             | Y                           | N             | Y           |
> > | Async TP (Micro-pipelining)           | Y             | Y                           | N             | Y           |
> > | CP                                    | Y             | Y                           | N             | N           |
> > | PP-Gpipe                              | Y             | Y                           | Y             | N           |
> > | PP-Interleaved (1F1B)                 | Y             | Y                           | Y             | Y           |
> > | PP-Looped-BFS                         | Y             | N                           | N             | N           |
> > | PP-1F1B                               | Y             | Y                           | Y             | Y           |
> > | PP-Flexible-Interleaved-1F1B          | Y             | N                           | N             | N           |
> > | PP-ZeroBubble                         | Y             | N                           | N             | Y           |
> > | (TP+SP)+PP                            | Y             | Y                           | N             | Y           |
> > | DDP+(TP+SP)+PP                        | Y             | Y                           | N             | Y           |
> > | FSDP +(TP + SP)                       | Y             | N                           | N             | N           |
> > | FSDP + (TP + SP) + PP                 | Y             | N                           | N             | N           |
> > | FSDP + (TP + SP) + PP + CP            | Y             | N                           | N             | N           |
> > | MoE                                   | Ongoing       | Y                           | N             | N           |
> > | Full AC                               | Y             | Y                           | Y             | Y           |
> > | Flexible SAC                          | Y             | N                           | N             | N           |
> > | DCP                                   | Y             | Y                           | Y             | Y           |
> > | Float-8 Training                      | Y             | Y                           | N             | N           |
> > | torch.compile                         | Y             | N (Custom Fusion Kernels)   | Partial       | N           |
> >
> > **Code Complexity:**
> >
> > These same design principles also contribute to LEGOSCALE’s reduced code complexity. Despite offering a rich feature set, LEGOSCALE has a compact and maintainable codebase, significantly smaller than other systems. Its modular architecture ensures extensibility and performance while minimizing complexity, making the framework easier to maintain and evolve.
> >
> > | **Lines of Code (LOC)**             | **Legoscale** | **Megatron-LM** | **DeepSpeed** |
> > |-------------------------------------|---------------|------------------|---------------|
> > | **Core Codebase**                   | 7K            | 93K             | 94K           |
> > | **Total Codebase (Including Utils)**| 9K            | 269K            | 194K          |
> >
> > LEGOSCALE achieves comparable or superior functionality with a fraction of the codebase, making it easier to maintain, extend, and integrate.

---

> ### Author Response · Authors · 2024-11-21
>
> **W3:**
>
> We thank the reviewer for their insightful observation. Although recent pre-trained LLM checkpoints primarily use FP16 or BF16, results with FP8 have been promising, and LEGOSCALE is designed to be future-ready. In LEGOSCALE, we adopt a cautious approach by applying FP8 selectively to linear layers, ensuring significant performance improvements while maintaining training stability.
>
> Our experiments show a 49% improvement in throughput with FP8 compared to BF16, alongside identical loss convergence curves. To demonstrate FP8’s efficacy, we compare training speed and loss convergence under equivalent configurations, highlighting its feasibility and benefits. While LEGOSCALE is primarily designed for pre-training, it also enables future exploration of fine-tuning on FP8-pretrained models.
>
> We will include detailed loss convergence curves in the appendix to support our findings further.
> Our graphs can be found here: [FP-8-Loss-Convergence](https://www.dropbox.com/scl/fo/w7pdixb5a8m8kyffycsxe/AMYZPLKvMutEfgPQSSiyDYE?rlkey=qm8i648i95b4s52eya722tej7&st=43aa066z&dl=0)
>
> **W4:**
>
> We thank the reviewer for their observation regarding LEGOSCALE’s use of PyTorch’s Distributed Checkpointing (DCP) and will include additional details about its design and performance in the appendix. LEGOSCALE addresses the challenge of production-critical features being dispersed across third-party libraries by building on the unified abstraction of **DTensor**, simplifying state dicts and enabling an efficient DCP implementation that is compatible across diverse parallelisms.
>
>  Comparison with Other Systems
>
> - **Gemini**: Tightly coupled with DeepSpeed, unavailable to native PyTorch users, supports only ZeRO-3 (FSDP), and lacks compatibility with TP, CP, and PP.
> - **ByteCheckpoint**: Adds asynchronous operations and disaggregated storage but relies on DCP’s foundation without significant enhancements for advanced parallelism.
>
>
> ### LEGOSCALE’s DCP Integration
>
> LEGOSCALE enhances DCP by integrating it with **DTensor** to standardize shard representations across TP, CP, PP, and FSDP. Asynchronous checkpointing overlaps storage operations with training, reducing overhead by **19x** compared to synchronous methods. A zero-overhead prototype further overlaps GPU-to-CPU transfers with forward and backward passes, achieving a **5x additional reduction** and reducing total checkpoint time to under one second. This unified, efficient design ensures LEGOSCALE’s checkpointing is robust, scalable, and production-ready for large-scale distributed training.
>
>
> Performance Graphs can be found here: [DCP-Performance](https://www.dropbox.com/scl/fo/w7pdixb5a8m8kyffycsxe/AMYZPLKvMutEfgPQSSiyDYE?rlkey=qm8i648i95b4s52eya722tej7&st=43aa066z&dl=0)
>
> **W5:**
>
> We acknowledge the reviewer’s observation that frameworks like JAX and ONNX leverage compiler optimizations to enhance training throughput and resource utilization. PyTorch and JAX, however, differ fundamentally in their programming paradigms:
>
> - **PyTorch** uses an imperative style with dynamic computation graphs, enabling immediate execution. This approach is intuitive, user-friendly, and facilitates debugging and fast prototyping, making it highly accessible.
> - **JAX**, in contrast, employs a functional paradigm with static computation graphs, achieving performance gains via XLA but introducing a steeper learning curve for users unfamiliar with functional programming.
>
> PyTorch’s imperative style, combined with its Pythonic syntax, active community, and versatility, has made it the leading framework for deep learning, with over 20K papers published in the last 12 months using PyTorch.
>
> LEGOSCALE bridges the gap between these paradigms by introducing **composability and modularity** within PyTorch’s imperative environment. This approach retains the ease of programming and fast prototyping of PyTorch while achieving performance benefits akin to compiler-driven frameworks like JAX.
>
> To address the reviewer’s point, `torch.compile`, introduced in PyTorch 2.0, enables compiler optimizations without compromising accessibility. LEGOSCALE enhances DTensor compatibility with `torch.compile`, allowing parallelism strategies like TP, CP, and FSDP to fully leverage these optimizations. This ensures state-of-the-art memory and computational efficiency while maintaining PyTorch’s flexibility. LEGOSCALE delivers a performant and modular system for pretraining large language models, balancing usability with significant efficiency gains.

---

> > ### Comment · Reviewer_3H4A · 2024-12-02
> >
> > I appreciate the authors' response. I have raised my score a bit.
> > Training LLMs with FP8 is not convincing with the reply.
> > The contribution of this manuscript is incremental.

---

> > > ### Author Response · Authors · 2024-12-02
> > > **FP8 Evals are super-promising**
> > >
> > > ## Effectiveness of FP8 Training
> > >
> > > Thank you for your feedback. We acknowledge that training LLMs with FP8 may not yet be fully convincing, and we appreciate the opportunity to clarify further.
> > >
> > > The experiments presented were conducted by one of our system’s users, not by the authors. They trained a 3B model following the Llama3 architecture on **1T tokens** using the **FineWeb-edu** dataset from Hugging Face. Evaluations were performed using the **lm-eval-harness** framework, and a portion of the results is presented below.
> > >
> > > While **bf16 performance is marginally better than float8 scores** (about 1%), some benchmarks show notable differences (e.g., **MMLU** is 3 points higher for bf16). These results are preliminary and were not optimized for hyperparameters or scaling. Notably, the bf16 run used **half the batch size**, which is known to improve evaluation scores. We expect these gaps to diminish with proper tuning and larger-scale training.
> > >
> > > | **Benchmark**     | **Score (float8)** | **Score (bf16)** |
> > > |--------------------|--------------------|------------------|
> > > | MMLU (5-shot)     | 0.26               | 0.29             |
> > > | ARC-e             | 0.73               | 0.73             |
> > > | ARC-c             | 0.43               | 0.46             |
> > > | Hellaswag         | 0.65               | 0.67             |
> > > | SciQ              | 0.89               | 0.88             |
> > > | OpenBook QA       | 0.43               | 0.43             |
> > > | PIQA              | 0.76               | 0.76             |
> > > | Winogrande        | 0.60               | 0.65             |
> > > | **Average**       | **0.59**           | **0.60**         |
> > >
> > > **Table:** Benchmark scores for float8-trained model evaluated in FP16 after **1T tokens** of FineWeb pre-training.
> > >
> > > We hope this provides additional context and emphasizes the early-stage nature of these results.

---

> > > > ### Comment · Reviewer_3H4A · 2024-12-02
> > > >
> > > > Was the optimizer also in FP8 in this experiment?

---

> ### Author Response · Authors · 2024-12-02
>
> Great question. No, the optimizer is still used in FP32. The optimizer states are sharded across all GPUs, so as the scale of training increases the cost of optimizer updates decreases linearly. For completeness, we use float8 linear layers, and float8 all gather in conjunction with FSDP2. For this training, we use float8 per tensor (tensorwise) scaling granularity rather than rowwise. We leverage torch.compile to ensure that we get maximum performance gains.

---

> > ### Comment · Reviewer_3H4A · 2024-12-02
> >
> > Can you share a bit more about FP16 in optimizer states? How was the stability issue addressed?

---

> > > ### Author Response · Authors · 2024-12-02
> > >
> > > Sorry, FP16 was a typo; perhaps you missed the edit to fix it before your response. The optimizer states are maintained in FP32.
> > >
> > > To recap the dtypes involved with float8 training with FSDP2:
> > > - The sharded parameters, gradients, and optimizer states are kept in FP32, as used by the optimizer step.
> > > - Conventionally, the FP32 sharded parameters are cast to BF16 before the all-gather, and for the float8 linear modules, the BF16 all-gathered weights are cast to float8 before computation. With float8 all-gather, the float8 linear weights’ casts to float8 are factored to before the all-gather to save communication bandwidth and shard the cast computation. Note that how FSDP2 supports mixing both float8 and BF16 parameters in the same all-gather kernel (by viewing the data as uint8 and handling the dtype conversions). Float8 is applied to the linears in the attention and MLP modules since they dominate the computation cost of LLM training, and their matmuls can leverage FP8 tensor cores in new GPUs (e.g. H100s).
> > > - The autograd-computed BF16 gradients are cast to FP32 and reduce-scattered to get the sharded gradients. This cast can be configured to happen after the reduce-scatter to save communication bandwidth, but we find FP32 reduce-scatter to be helpful for convergence, especially for larger world sizes.
> > >
> > > LEGOSCALE offers this kind of flexibility in dtypes that enables exploration in float8 training.

---

> > > > ### Comment · Reviewer_3H4A · 2024-12-02
> > > >
> > > > Appreciate it. As a tool, I would recommend LEGOSCALE for acceptance.

---

> > > > > ### Author Response · Authors · 2024-12-03
> > > > >
> > > > > Thank you for the discussion. We're glad to receive your recommendation and hope that your decision will be reflected in our final score.

---

### Comment · Area_Chair_dNBq · 2024-11-23
**Engage in Discussions Before Nov 26 (AoE)**

Dear Reviewers,

First, let me thank you for your invaluable contributions to the ICLR review process. Your constructive feedback plays a key role in enhancing the quality of submissions.

---

As we approach the final days of the discussion phase (ending **Nov 26, 2024, AoE**), I kindly remind you to:

- Please take a moment to review the authors' responses to your comments. This is an opportunity to clarify any remaining questions, acknowledge misunderstandings, and refine your evaluation.

- If you need further clarification, don't hesitate to post your comments as soon as possible.

- If the authors' responses address your concerns or provide new insights, please consider updating your score to reflect this.

---

Your thoughtful participation during this phase is especially valuable for borderline papers, where additional input can be critical to ensuring a fair decision-making process.

I understand how busy this time of year can be and truly appreciate the time and care you dedicate to this important role. Your efforts make a tangible impact on the success of ICLR.

Thank you once again for your dedication.

Best regards,

Area Chair, ICLR 2025

---

### Author Response · Authors · 2024-11-28
**Revision Summary (Updated Manuscript)**

We thank the reviewers for appreciating our work, recognizing its impact, highlighting its strengths, asking meaningful questions, and providing valuable feedback. To address the reviewers' concerns, we have revised our manuscript. **All changes made are highlighted in blue**.

Below, we provide a comprehensive list of changes addressing the reviewers' questions and concerns. For clarity, we denote:
- **Ri**: Reviewer `i`
- **Qj**: Question `j`
- **Wk**: Weakness `k`

## List of Changes

1. **Novelty and Research Contributions** [R1W1, R2W1, R4W2, R4Q2, R5W1, R6Q0]
   - Clearly explained the novelty and research contributions of LEGOSCALE in the introduction (**Section 1**).

2. **Significant Advancements and Comparison to Existing Systems** [R1W2, R4Q2, R5W1, R6Q1, R6Q2]
   - Enhanced the related work section (**Section 5**) and provided a detailed feature comparison across systems demonstrating the significant contributions of LEGOSCALE (**Appendix B9, Table 6**).

3. **Float8 Training Loss Convergence** [R1W3, R4W3]
   - Included loss convergence graphs for Float8 training on an AWS cluster with 32 H100 GPUs (**Appendix B11, Figure 7**).

4. **Incorporating Context Parallelism (CP) for 4D Parallelism** [R2W2, R3W1, R6W3]
   - Demonstrated how to extend and incorporate CP for 4D parallel training and large context scaling (300k using CP degree 8) (**Section 4.1** and **Appendix B8**).

5. **FSDP1 vs FSDP2** [R2Q1, R2W4]
   - Highlighted the pointer in the main text to the appendix summarizing FSDP2 vs. FSDP1 design changes (**Appendix B1**).

6. **Loss Convergence and Correctness** [R3W3, R4W3, R5Q2]
   - Showed loss convergence graphs for Llama 8B, 70B, and 405B for 1D and 2D training over 3k epochs (**Appendix B10, Figures 5 and 6**).

7. **Metrics Clarification in Table Captions** [R3Q1]
   - Improved table captions for better readability.

8. **Additional Experimental Details on ZeRO Configurations** [R3W4, R3Q4]
   - Provided additional experimental details (**Appendix B10**).

9. **HSDP in LEGOSCALE** [R3Q5]
   - Highlighted the pointer in the main text to existing details in the appendix (**Appendix B2**).

10. **Improved Figure Readability** [R4W4]
    - Magnified figures for better clarity.

---

## To the Reviewers

We have addressed all your questions and weaknesses and incorporated the suggested changes into our manuscript. We kindly request you to review them and provide any additional comments or questions you may have. Your feedback is invaluable.


## Reviewer Mapping
- **R1**: Reviewer 3H4A
- **R2**: Reviewer izCA
- **R3**: Reviewer Xmiy
- **R4**: Reviewer 5X5L
- **R5**: Reviewer DprR
- **R6**: Reviewer UCk4

---

### Meta-Review · Area_Chair_dNBq · 2024-12-23

**Metareview:**

The paper introduces LegoScale, an open-source, PyTorch-native distributed training framework for LLM pre-training. The framework unifies SOTA techniques for distributed training, including FSDP, TP, PP, CP, and different numerics and activation checkpointing. LegoScale provides a modular, composable, and extensible abstractions that aim to facilitate integration and reduce the engineering overhead of pre-training LLMs.

The paper has the following strengths: (a) addresses a critical need in LLM pre-training by democratizing access to scalable distributed training methods, which are often fragmented across different frameworks. (b) from engineering perspective, the paper includes a significant engineering effort, integrating advanced parallelism techniques with PyTorch's distributed library while adhering to good software engineering methodology. (c) the evaluation includes benchmarks on Llama 3 models, showing training speedups and effective scaling across hardware and model dimensions. (d) open-source framework, which seems has garnered interest from the community already. (e) principled design in which LegoScale allows seamless integration of new parallelism strategies and optimizations.

On the other hand, the paper has some weaknesses as well: (a) from research perspective, the paper may sound purely an engineering effort (which is not necessarily an issue for publication) that consolidates existing techniques. (b) some reviewers expressed concerns about the long-term relevance of the work, especially because of the rapid evolution of PyTorch and distributed training systems. (c) some reviewers also mentioned that because the paper attempts to cover too many modules, it makes it less accessible to readers without ML system background.

After reading the reviewers' comments, the authors' rebuttal, and the discussion, I recommend **Accept** for the following reasons:

(a) This is a production-grade framework that covers a wide range of parallelism method and can be useful for the ML community. The ability of the framework to unify (or at least attempt to unify) and improve distributed training workflows is likely to have significant impact, particularly for researchers and practitioners working on LLMs.

(b) The open-source nature of this work and already active community engagement supports the value of the framework.

(c) The rebuttal addressed most of the reviewer concerns, including clarifications on contributions, comparisons with related systems, and update to incorporate CP and expert parallelism (I believe the community can even further extend the framework).

(d) While I agree with the reviewers regarding limited research novelty, the detailed design of LegoScale can be a strong research tool (and possibly a strong baseline for comparisons) for the community and can inspire further innovations and extensions in the field.

Overall, this submission represents a valuable addition to the ML community, providing a well-engineered solution to challenges in LLM pre-training and serving as a benchmark for distributed training frameworks.

**Additional Comments On Reviewer Discussion:**

## Reviewers Comments

Overall all the reviewers were positive about the paper.

- Several reviewers (3H4A, izCA, DprR) questioned the novelty of LegoScale, emphasizing that the work primarily consolidates existing techniques without introducing significant methodological innovation.
- Reviewers highlighted the importance of clearly distinguishing LegoScale’s unique contributions from prior frameworks like DeepSpeed, Megatron-LM, and veScale.
- Reviewers (Xmiy, UCk4) requested results on broader scenarios such as context scaling, expert parallelism, and multimodal models.
- Some reviewers (DprR) questioned the longevity of LegoScale, given the evolving capabilities of PyTorch and competing frameworks. They asked about the authors’ plans for long-term maintenance and support.
- Reviewer (UCk4) also mentioned that the paper is dense and less accessible to readers without an ML systems background and some figures hard to read.

---

## Authors' Responses

- The authors emphasized that LegoScale’s primary contribution lies in its unified design principles, which enable seamless integration of diverse parallelism techniques and optimizations.
- The authors provided detailed comparisons with related frameworks and clarified LegoScale’s distinct advantages, such as its broader feature coverage, reduced code complexity, and flexibility.
- The authors included additional experiments, such as loss convergence curves for Float8 training, ablation studies on FSDP1 vs. FSDP2, and scaling results for Context Parallelism (CP) -- which I strongly believe should be added to the appendix of the paper.
- The authors highlighted significant community adoption (GitHub stars and active contributors), and extensive usage across academic and industry clusters.
- Per reviewers' suggestions, CP and expert parallelism were integrated into LegoScale during the rebuttal period, addressing reviewer concerns about scalability and support for broader scenarios.

Based on the discussion, Reviewer (DprR) also increased their scores and acknowledged the potential impact on the community.

---

### Decision · Program_Chairs · 2025-01-22

Accept (Poster)